

# Method to retrieve cloud condensation nuclei number concentrations using lidar measurements

Wangshu Tan[1], Gang Zhao[1], Yingli Yu[1], Chengcai Li[1], Jian Li[2], Ling Kang[3], Tong Zhu[3], Chunsheng Zhao[1]

[1]Department of Atmospheric and Oceanic Sciences, School of Physics, Peking University, Beijing 100871, China
[2]State Key Laboratory of Severe Weather & Key Laboratory of Atmospheric Chemistry of CMA, Chinese Academy of Meteorological Sciences, Beijing 100081, China
[3]State Key Joint Laboratory of Environmental Simulation and Pollution Control, College of Environmental Science & Engineering, Peking University, Beijing 100871, China

*Correspondence to*: Chengcai Li (ccli@pku.edu.cn)

**Abstract.** Determination of cloud condensation nuclei (CCN) number concentrations at cloud base is important to constrain aerosol-cloud interactions. A new method to retrieve CCN number concentrations using backscatter and extinction profiles from multiwavelength Raman lidars is proposed. The method implements hygroscopic enhancements of backscatter/extinction with relative humidity to derive dry backscatter/extinction and humidogram parameters. Humidogram parameters, Ångström

exponents, and lidar extinction-to-backscatter ratios are then linked to the ratio of CCN number concentration to dry backscatter/extinction coefficient ($AR_\xi$). This linkage is established based on the datasets simulated by Mie theory and $\kappa$-Köhler theory with in situ measured particle size distributions and chemical compositions. CCN number concentration can thus be calculated with $AR_\xi$ and dry backscatter/extinction. An independent theoretical simulated datasets is used to validate this new method and results show that the retrieved CCN number concentrations at supersaturations of 0.07%, 0.10%, and

0.20% are in good agreement with theoretical calculated values. Sensitivity tests indicate that retrieval error in CCN arise mostly from uncertainties in extinction coefficients and RH profiles. The proposed method improves CCN retrieval from lidar measurements and has great potential in deriving scarce long-term CCN data at cloud base which benefits aerosol-cloud interaction studies.

## 1 Introduction

Anthropogenic activities have caused a huge increase in atmospheric aerosols, and some of the aerosol particles affect the climate by serving as cloud condensation nuclei (CCN). CCN in clouds can modify cloud forming processes and cloud microphysical properties (Rosenfeld et al., 2014). Although numerous impacts of aerosol-cloud interactions on radiative forcing (McCoy et al., 2017;Zhou et al., 2017), precipitation (Xu et al., 2017;Fan et al., 2018), cloud electrification (Wang et al., 2018), and severe weathers or hazards (Fu et al., 2017) have been discovered, constraining the relationships between



aerosols and clouds is still a big challenge (Seinfeld et al., 2016). Lacking the knowledge of aerosol-cloud interactions limits our ability to estimate climate forcing caused by aerosols (Boucher et al., 2013).

Aerosol CCN supersaturation activation spectrum is one of the most critical parameters to quantify aerosol-cloud interactions (Schmale et al., 2018). Despite that a large amount of CCN number concentrations near ground have been measured worldwide (Tao et al., 2017), ground-measured CCN may not represent CCN at cloud base that alter clouds directly. Obtaining CCN near cloud base becomes a crucial issue. Cloud base CCN can be measured in situ on aircraft platforms, but airborne measurements have the limitations of huge costs and discontinuity. Satellites are difficult to observe CCN at cloud base, because clouds always obscure aerosol signals beneath them. Rosenfeld et al. (2016) have proposed an alternative approach for satellites to retrieve CCN concentrations using clouds as CCN chambers, however, employing CCN concentrations derived with this strategy limits our exploration of the relationship between CCN concentrations and cloud droplet concentrations in natural environment. So far, CCN concentrations at cloud base are scarce for aerosol-cloud interaction studies.

Ground-based lidars can continuously provide optical properties of aerosol particles from ground up to cloud base (Mattis et al., 2016;Li et al., 2019), showing great potential in deriving CCN concentrations near cloud base. Ghan and Collins (2004) propose a simple method to infer CCN profiles with the combination of surface in situ CCN and aerosol optical measurements. The method is only applicable when boundary layer is well mixed from surface to cloud base (Ghan et al., 2006). Multiwavelength Raman lidars (MWRLs) are increasingly used to detect aerosol vertical distributions in recent years. The principle of MWRLs allows independent retrieval of particle backscatter ($\beta$) and extinction coefficients ($\alpha$), which provides more information about particle microphysical properties (Müller et al., 2016). Existing approach to retrieve CCN using MWRLs is based on microphysical inversion techniques (Mamouri and Ansmann, 2016). Retrieved optical-equivalent particle size distributions together with assumed activation critical diameters are utilized to calculate CCN concentrations (Lv et al., 2018).

There are three major challenges in CCN concentration retrieval with lidars. The first is the conversion of lidar-derived optical properties into particle number concentrations. High uncertainties of retrieved particle number concentrations could be an important source of CCN retrieval error. The second one is the determination of particle hygroscopicity in order to evaluate the ability of particles to participate as CCN. Particle hygroscopicity, which is highly related to chemical composition and aging/coating effect, is found to cause nonnegligible variations in cloud droplet activation (Hudson, 2007;Zhang et al., 2017). The last is the influence of high relative humidity (RH) near clouds. Aerosol particles are likely to be humidified in ambient environment, and the consequent changes in optical properties make CCN retrieval more complicated. Most studies working on CCN retrieval with MWRLs mainly focus on deriving particle number concentrations, but seldom commence to solve the issue of hygroscopicity.

In recent years, several aerosol hygroscopic studies based on lidar measurements have been carried out (Fernández et al., 2017;Lv et al., 2017;Bedoya-Velásquez et al., 2018). Backscatter and extinction enhancement factors can be derived with lidar measurements and RH profiles. The enhancement factor, which is associated with both particle size and hygroscopicity (Kuang et al., 2017), is defined as:



$$f_\xi(\text{RH}, \lambda) = \frac{\xi(\text{RH}, \lambda)}{\xi(\text{RH}_{\text{ref}}, \lambda)}, \tag{1}$$

where $f_\xi$ is the enhancement factor of the optical property $\xi$ (backscatter or extinction) at a specific light wavelength $\lambda$ and RH, and $\text{RH}_{\text{ref}}$ is the reference RH value. Many studies manifest that lidar-derived enhancement factors are in good agreement with in situ measurements (Wulfmeyer and Feingold, 2000;Pahlow et al., 2006;Fernández et al., 2015;Rosati et al., 2016).

Feingold and Morley (2003) demonstrate that the extent of backscatter and extinction enhancements hints the ability of particles to serve as CCN. Tao et al. (2018) use in situ measured light scattering enhancement factors to predict $N_{\text{CCN}}$ at 0.07% supersaturation, and the result shows strong consistency with CCN counter.

In this paper, a new method to retrieve CCN number concentrations for $3\beta+2\alpha$ MWRL systems (backscatter coefficients at 355, 532, and 1064 nm and extinction coefficients at 355 and 532 nm) is proposed based on $\kappa$-Köhler theory (Petters and

Kreidenweis, 2007) and Mie theory (Bohren and Huffman, 2007). Enhancements of backscatter and extinction with RH are first implemented in CCN retrieval using MWRLs. The paper is structured as follows. Section 2 introduces the measured and simulated datasets used in this paper. Section 3 presents the methodology. Firstly, suitable supersaturation conditions for lidar retrieval are discussed in Sect 3.1. Performances of two parameterization scheme for backscatter and extinction humidogram are evaluated in Sect 3.2. In Sect. 3.3, the new CCN retrieval method for MWRLs are described in detail. Sensitivity tests are

carried out in Sect. 3.4. Results and summary are given in Sect. 4 and Sect. 5, respectively.

## 2 Data

### 2.1 Datasets of aerosol properties

In situ measured aerosol properties were collected from five field campaigns at three different measurement sites in the North China Plain (NCP). The measurement sites are located at Wuqing (39°23′ N, 117°01′ E, 7.4 m a.s.l) in Tianjin, Xianghe (39°45′

20  N, 116°58′ E, 36 m a.s.l) and Wangdu (38°40′ N, 115°08′ E, 51 m a.s.l) in Hebei province. The specific locations, topographical information, and pollution status of these measurements sites are shown in Fig. S1 in the Supplement. These three sites all lie inside the polluted NCP region and are highly representative of the polluted background (Xu et al., 2011;Bian et al., 2018;Sun et al., 2018). Time periods, measured parameters, and corresponding instruments of individual campaign are listed in Table 1. During these field campaigns, except measurement for size-resolved chemical compositions, ambient particles were drawn in

through a PM10 inlet (16.67 L/min), passed through a silica gel diffusion drier, and then were split into different instruments. All instruments were operated at RH less than 30%.

The particle number size distributions (PNSDs) were measured with the combination of a twin differential mobility particle sizer (TDMPS, IfT, Leipzig, Germany) or a scanning mobility particle size spectrometer (SMPS) and an aerodynamic particle sizer (APS, TSI, Inc., Shoreview, MN USA, Model 3320 or Model 3321). The statistical information about the measured

PNSDs is shown in Fig. 1a. The peaks of the PNSDs are at about 100 nm (diameter in log-scale), which shows strong characteristics of continental aerosols.



The black carbon (BC) mass concentrations ($m_{BC}$) were measured by a multi-angle absorption photometer (MAAP, Thermo, Inc., Waltham, MA USA, Model 5012). As for mixing states of BC, BC and other non-absorbing compositions were found to be both externally mixed and core-shell mixed during the campaigns (Ma et al., 2012). The mass fraction of externally-mixed BC ($r_{ext}$) is defined to quantify the mixing states of BC:

$$r_{ext} = \frac{m_{ext\_BC}}{m_{BC}},$$ (2)

where $m_{ext\_BC}$ is the mass concentration of externally mixed BC. According to Ma et al. (2012), $r_{ext}$ can be retrieved from hemispheric backscattering fractions (HBFs) measured by an integrating nephelometer (TSI, Inc., Shoreview, MN USA, Model 3563).

Size-resolved chemical compositions all come from campaign C2. The size-resolved aerosol sampling was carried out with a ten-stage Berner low pressure impactor (BLPI). Chemical species including inorganic ions ($NH_4^+$, $Na^+$, $K^+$, $Mg^{2+}$, $Ca^{2+}$, $NO_3^-$, $SO_4^{2-}$, $Cl^-$), elemental carbon, organic carbon, water-soluble organic carbon and some other species such as dicarboxylic acids were analyzed from sample substrates. After transforming the ambient wet aerodynamic diameters into dry volume-equivalent diameters, size-resolved $\kappa$ distributions were derived from measured size-resolved chemical compositions. Twenty-five typical size-resolved $\kappa$ distributions in the NCP are given in Fig. 1b. The measured size-resolved $\kappa$ distributions vary a lot and cover a wide range of aerosol hygroscopicity (Kuang et al., 2018). More details about the measurements can be found in Liu et al. (2014).

## 2.2 Datasets of CCN number concentrations and lidar-derived optical properties

In situ measured aerosol properties mentioned above are utilized to calculate CCN number concentrations and particle backscatter and extinction coefficients base on $\kappa$-Köhler theory and Mie theory. For each simultaneously measured PNSD, $m_{BC}$, and $r_{ext}$ (16183 sets of data), simulations are carried out with every one of the twenty-five size-resolved $\kappa$ distributions. Petters and Kreidenweis (2007) introduce the $\kappa$-Köhler equation to describe the relationship between particle/droplet diameter $D$ and critical supersaturation ratio (SS) or RH with a single hygroscopic parameter $\kappa$:

$$RH(D) = 1 + SS(D) = \frac{D^3 - D_{dry}^3}{D^3 - D_{dry}^3(1-\kappa)} \exp\left(\frac{4\sigma_{s/a}M_w}{RT\rho_w D}\right),$$ (3)

where $D_{dry}$ is particle dry diameter, $\sigma_{s/a}$ is the surface tension of the solution/air interface, $M_w$ is the molecular weight of water, $R$ is the universal gas constant, $T$ is temperature, and $\rho_w$ is the density of water. Given PNSD at dry condition, SRKD, and $r_{ext}$, $\kappa$-Köhler equation can be used to estimate CCN number concentrations by calculating critical diameter. CCN number concentrations at the supersaturations of 0.07%, 0.10%, 0.20%, 0.40%, and 0.80% are accordingly simulated. The selected supersaturation ratios are widely used in CCN measurements.

Mie theory can solve light scattering problems of homogeneous and coated spherical particles. Without the consideration of mineral dust, using Mie model is quite reasonable because particles are likely to be spherical near clouds where the RH cloud be relatively high. When simulating particle backscatter and coefficients, PNSD, $m_{BC}$, $r_{ext}$, and complex refractive index are





essential. PNSD at different RH can be calculated with $\kappa$-Köhler equation as well. The refractive indices of BC, non-absorbing component, and pure water are set to be $1.8+0.54i$ (Ma et al., 2012), $1.53+10^{-7}i$ (Wex et al., 2002), and $1.33+10^{-7}i$ respectively. More detail about calculations of lidar-derived optical properties can be found in (Zhao et al., 2017). Backscatter coefficients (355, 532, and 1064 nm) and extinction coefficients (355 and 532 nm) at dry condition and RH from 60-90% are simulated

with an interval of 1%.

## 3 Methodology

### 3.1 Supersaturations for lidar CCN retrieval

CCN number concentrations are related with supersaturations. Critical diameters of each supersaturations calculated with twenty-five size-resolved $\kappa$ distributions are shown in Fig. 2a. Most of the critical diameters at supersaturation of 0.07% are

larger than 200 nm, while critical diameters at supersaturation of 0.80% are around 50 nm. Suitable supersaturations for lidar CCN retrieval depend on the ability of lidar optical properties to provide information about number and hygroscopicity of CCN-related sizes.

Size cumulative contributions of particle number of all measured particle size distribution and corresponding calculated backscatter and extinction at dry condition are also displayed in Fig. 2a. As the cumulative contributions of particle number

suggest, particles with diameter less than 100 nm dominate particle number concentrations (over 65%). However, most backscatter and extinction come from particles larger than 200 nm (around 90%) and almost 100% come from particles larger than 100 nm. If critical diameter is small, dry backscatter and extinction are insensitive to particles diameters that contribute to most CCN concentrations.

Size-resolved enhancement contributions of backscatter and extinction are calculated to discuss hygroscopicity sensitive size

of optical enhancement factor measurement. The enhancement contribution is defined as the difference between optical cross-sections of RH at 90% and 60%, and represents the proportion of each size to the enhancement in backscatter or extinction. As is shown in Fig. 2b, the contributions of the extinction enhancements are concentrated in the diameters within 200 nm to 700 nm, and extinction enhancement at 355 nm is related to smaller particles than that at 532 nm. Strong oscillations are found in size enhancement contributions of backscatter coefficients. Similar to particle number, particles with diameters smaller than

100nm contributes little to the enhancements of both backscatter and extinction.

Comparing sensitive size of optical properties and critical diameters at different supersaturations. $3\beta+2\alpha$ MWRL systems have potential to retrieve CCN number concentrations at supersaturations smaller than 0.20%. It is not recommended to estimate CCN concentrations using lidar data at superstations larger than 0.40%.

### 3.2 Humidogram parameterization for backscatter and extinction enhancements

Humidogram parameterization is needed to find a representative parameter for the relationship between enhancement factor and RH. Unlike in situ controlled-RH measurements, there is no such a generic reference RH as dry condition for lidar



measurements to derive enhancement factor. Inferring backscatter and extinction coefficients at dry condition ($\xi_{dry}$) is also an important issue in CCN retrieval. Therefore, humidogram parameterization of lidar-derived optical properties should combine $\xi_{dry}$ and $f_\xi(RH, \lambda)$ together.

Many equations to parameterize enhancement factors have been proposed by previous studies (Titos et al., 2016). Two one-

parameter equations are selected to test their performance on estimating $\xi_{dry}$ and representing particle hygroscopic growth characteristics. The first equation is the most commonly used one initially introduced by Kasten (1969):

$$\xi(RH, \lambda) = \xi_{dry}(\lambda) \cdot f_\xi(RH, \lambda) = \xi_{dry}(\lambda) \cdot (1 - RH)^{-\gamma_\xi(\lambda)} , \qquad (4)$$

where the exponent $\gamma_\xi$ is the fitting parameter and describes the hygroscopic behavior of the particles; the other equation is proposed based on physical understanding by Brock et al. (2016), which has been reported to have better performance in

describing light scattering enhancement factor than Eq. (4) (Yu et al., 2018):

$$\xi(RH, \lambda) = \xi_{dry}(\lambda) \cdot f_\xi(RH, \lambda) = \xi_{dry}(\lambda) \cdot \left[1 + \kappa_\xi(\lambda) \frac{RH}{1-RH}\right], \qquad (5)$$

where $\kappa_\xi$ is the fitting parameter and shows significant correlation with bulk hygroscopic parameter $\kappa$ (Kuang et al., 2017). Here, Eq. (4) and Eq. (5) are denoted as $\gamma$-equation and $\kappa$-equation respectively. With given backscatter and extinction at different RH, $\xi_{dry}$ and $\gamma_\xi$ or $\kappa_\xi$ can be fitted simultaneously by means of least squares.

Comparisons between the performances of $\gamma$-equation and $\kappa$-equation on inferring backscatter and extinction at dry condition are carried out to select a better parameterization. Four RH ranges (60%-90%, 60%-70%, 70%-80%, and 80%-90%) are selected. The fitted $\xi_{dry}$ are compared with the $\xi_{dry}$ calculated by Mie model. The slopes of linear regressions, determination coefficients ($R^2$), and relative errors are listed in Table 2. Apparently, $\kappa$-equation has a better performance than $\gamma$-equation for all RH ranges. Inferring $\xi_{dry}$ with $\gamma$-equation will underestimate about 10%-30%. It is consistent with the finding of Haarig et

al. (2017) that $\gamma$-equation does not hold for RH lower than 40%. The bias of backscatter is found to be larger than the bias of extinction.

The RH range of humidogram equations also influences the fitting results. Table 2 shows the fitted $\xi_{dry}$ have larger bias when the value of RH increase. The fitted humidogram parameters $\gamma_\xi$ and $\kappa_\xi$ from different RH ranges are compared to each other, and the results are displayed in Table 3. Parameterization equations are not always perfect for the whole RH ranges, so

humidogram parameters fitted with various RH ranges can be different. If $\gamma_\xi$ and $\kappa_\xi$ are used to represent hygroscopic behavior of particles, more careful attention should be paid to the RH ranges.

Based on the comparisons above, Eq. (5) ($\kappa$-equation) is selected as our humidogram equation to derive $\xi_{dry}$ and $\kappa_\xi$. The RH range for parameter fitting used is fixed to 60%-90% in the following method.



### 3.3 Method to retrieve CCN number concentrations using MWRL

#### 3.3.1 Overview

An optical-related CCN activation ratio, $AR_\xi$, is introduced to bridge the gap between CCN and lidar-derived optical properties. $AR_\xi$ is the ratio between CCN number concentration and backscatter/extinction coefficient, which can be expressed

as:

$$AR_\xi(SS, \lambda) = \frac{N_{CCN}(SS)}{\xi_{dry}(\lambda)} = \frac{N_{CCN}(SS)}{N_{aerosol}} \cdot \frac{N_{aerosol}}{\xi_{dry}(\lambda)}, \tag{6}$$

where $N_{CCN}$ is the CCN number concentration, and $N_{aerosol}$ is the total number concentration of aerosol particles. $AR_\xi$ can be divided into two parts: one is the ratio of CCN to the total particles, which is the origin definition of CCN activation ratio; the other is the ratio of total number concentration to backscatter or extinction at dry condition. Bulk CCN activation ratio is

related with particle size distribution and hygroscopicity, and the relationship between particle number concentration and optical properties is mainly controlled by size distribution. Therefore, $AR_\xi$ could be quantified with size and hygroscopicity information.

With the method in Sect. 3.2, $\xi_{dry}$ and $\kappa_\xi$ can be derived with backscatter and extinction enhancements. Optical humidogram parameters $\kappa_\xi$ can be regarded as parameters indicating hygroscopicity. Extinction-related Ångström exponent ($\mathring{a}_\alpha$) is the most

commonly used parameter to reveal information about the predominant size of aerosols. Generally speaking, a smaller $\mathring{a}_\alpha$ represents there are more large particles. Similarly, backscatter-related Ångström exponent ($\mathring{a}_\beta$) are often employed in lidar analysis (Fernández et al., 2015), and particle backscatter coefficients of different wavelengths also have been proved to have a valid Ångström exponent relationship. Ångström exponent of dry backscatter and extinction coefficients ($\mathring{a}_\xi$) between two wavelengths can be derived using Eq. (7):

$$\mathring{a}_\xi(\lambda_1, \lambda_2) = -\frac{\log(\xi_1/\xi_2)}{\log(\lambda_1/\lambda_2)}, \tag{7}$$

where the subscript 1 and 2 represents different wavelengths. Another widely used parameter to express aerosol characteristics in lidar studies is the particle lidar extinction-to-backscatter ratio (lidar ratio, $s_a$), which is defined as the ratio of extinction coefficient to backscatter coefficient at a specific light wavelength:

$$s_a(\lambda) = \frac{\alpha(\lambda)}{\beta(\lambda)} = \frac{4\pi}{P(\pi) \cdot \omega}. \tag{8}$$

As is shown in Eq. (7), lidar ratio is determined by the scattering phase function at $180°$ $P(\pi)$ and the single scattering albedo $\omega$. $P(\pi)$ is mainly influenced by particle size and $\omega$ indicates the content and mixing state of light absorbing components. Lidar ratio is often utilized in aerosol type classification and is proved to be very sensitive to particle sizes (Zhao et al., 2017). Particle type information can also be regard as an alternative representative of hygroscopicity. Therefore, lidar ratio of dry particles could be a reliable parameter to estimate $AR_\xi$.

Statistical relationship among $\mathring{a}_\xi$, $s_a$, $\kappa_\xi$, and $AR_\xi$ are used in our new method. Based on the statistical relationship, $AR_\xi$ can be estimated by $\mathring{a}_\xi$, $s_a$, and $\kappa_\xi$. The implement of $\mathring{a}_\xi$ and $s_a$ is quite similar to the microphysical inversion process for particle



size distribution retrieval. Microphysical inversion is a physics-based approach but will bring huge uncertainties in retrieving particle number concentrations. Constraining $AR_\xi$ directly with statistical relationship is a much more simple and straightforward way.

After $AR_\xi$ of backscatter and extinction at different wavelengths are derived, CCN number concentration can be calculated by

multiplying $AR_\xi$ by the corresponding $\xi_{dry}$. The average value of CCN concentrations calculated by different $\xi_{dry}$ is the final retrieval result. The schematic diagram of the whole algorithm is shown in Fig. 3.

### 3.3.2 Appropriate retrieval layers

A constraint needs to be satisfied when quantifying the enhancements of backscatter and extinction coefficients with lidar measurements. The selected vertical layers must be well-mixed, so we can guarantee that the variations of particle

backscatter/extinction coefficients are caused by different RH and not by various aerosol types or loads. Atmospheric vertical homogeneity is fulfilled if the layer has little variability of virtual potential temperature profile and water vapor mixing ratio profile (Lv et al., 2017). Additional analyses can also be considered to evaluate vertical mixing of air masses, such as backward trajectory, horizontal wind velocities at different altitude, or the third moment of the frequency distribution of vertical wind velocities (Bedoya-Velásquez et al., 2018).

Once vertical homogeneity is ensured, physical and chemical properties at dry condition can be assumed to be uniform in the selected layer, and the number concentrations are proportional to air molecule number density. Accordingly, the relative variations of particle backscatter/extinction coefficients against different RH can be achieved after normalizing the backscatter and extinction coefficients with air molecule number density.

### 3.3.3 Estimation of $AR_\xi$

Ångström exponents, lidar ratios, and optical humidogram parameters $\kappa_\xi$ are used to estimate optical-related activation ratio $AR_\xi$. Concerning the Ångström exponents and lidar ratios are not independent to each other (any parameter can be calculated from other parameters), we reduce the number of parameters to a sufficient number to represent all the information. The selected nine parameters are listed in Table 4. There are no explicit expressions between these parameters and $AR_\xi$, and the relationships between them are highly nonlinear. One possible way to solve this problem is to build a lookup table, but too

many input parameters would make the lookup table so large to build and operate.

In the past few decades, machine learning has been a field that develops rapidly, which experiences a very wide range of applications (Grange et al., 2018). Compared to traditional statistical methods, many machine learning techniques are nonparametric and do not need to fulfill many assumptions required for statistical methods (Immitzer et al., 2012). Random forest (RF) is an ensemble decision tree machine learning method that can be used for regression. (Breiman, 2001;Tong et al.,

2003). Beside the free restraints on input parameters and assumptions, RF also has the advantage of being able to explain and investigate the learning process (Kotsiantis, 2013). The Python module *RandomForestRegressor* from the Python Scikit-Learn





library (http://scikit-learn.org/stable/modules/generated/sklearn.ensemble.RandomForestRegressor.html, last access: 18 December 2018) are utilized as the RF model.

Some tuning parameters required by RF model need to be specified by users. Experiments are made to determine the optimal values of the tuning parameters. Experiment results are showed in Fig. S3 in the Supplement and the detailed settings of the

RF model are listed in Table S1 in the Supplement. In this case, the results are rather insensitive to the tuning parameters. Data simulated with datasets measured from campaign C1-C4 are utilized as the training data, and those from C5 are used as test data.

### 3.4 Sensitivity test

Both systematic and random errors exist in lidar-retrieved backscatter and extinction coefficients (Mattis et al., 2016).

Systematic errors in backscatter and extinction can come from instrumentation setup, data processing method, and retrieval algorithm. Sensitivity test is carried out to test the impact of systematic errors of backscatter and extinction on CCN retrieval. Errors in backscatter or extinction influence the value of Ångström exponents and lidar ratios but have no impact on $\kappa_\xi$. The errors of individual backscatter or extinction are consider to be independent, though systematic errors of different parameters are related. The systematic errors are given in the range of -20% to 20% with an interval of 2%. In each test, the error is only

applied to one parameter, and other parameters are error-free.

RH is another crucial factor in this new method to retrieve CCN. Profiles of RH derived by remote sensing techniques are also influenced by errors. At present, RH profiles are usually obtained with the combination of temperature from microwave radiometer and water vapor mixing ratio from MWRL. Both measurements can cause systematic and random errors in RH (Bedoya-Velásquez et al., 2018). Errors in RH will influence the values of $\xi_{dry}$ and $\kappa_\xi$, which in turn influence all the nine

input parameters. Systematic errors ranging from -10% to 10% in intervals of 1% are considered for RH.

Random errors in observations can be reduced by temporal averaging but cannot be eliminated. The influence of random errors in backscatter, extinction, and RH on CCN retrieval are investigated with Monte Carlo method. Errors obeying Gaussian distribution are generated randomly with the mean value of zero. The standard deviation of Gaussian distribution is 10% for backscatter/extinction and 5% for RH. The procedure is repeated for 2000 times. All the 80575 sets of data from campaign C5

are used for sensitivity test.

## 4 Results

### 4.1 CCN number concentrations retrieved with error-free data

With error-free data as input, the model predicted extinction-related activation ratio at 532 nm ($AR_{\alpha532}$) and the retrieved CCN number concentrations at supersaturations of 0.07%, 0.10%, and 0.20% are compared to the theoretical calculated values. A

total of 80575 pairs of data calculated from campaign C5 are used for verification. The retrieval results are displayed in Fig.





4. The values $AR_{\alpha532}$ at a specific supersaturation are distributed in a wide range and can span over an order of magnitude, indicating that the relationship between CCN and optical parameters is very complex. According to Fig.4, all data points are distributed almost evenly on both sides of the 1:1 line and the relative errors of most points are within 20%. The determination coefficients ($R^2$) of CCN concentrations are all larger than 0.97, and the results do not show obvious systematic deviations.

The retrieval errors are found to grow with supersaturation. Retrieval results for higher supersaturations (i.e. 0.40% and 0.80%) is displayed in Fig. S4 in the Supplement. There are larger errors for supersaturations of 0.40% and 0.80%. Only 47.76% of the retrieved CCN number concentration at supersaturation of 0.80% have relative errors less than 20%. The results are consistent with the previous analysis in Sect. 3.1, which means lidars may not sufficient enough to retrieve CCN number concentrations at supersaturations lager than 0.40%.

**4.2 Importance of size-related and hygroscopicity-related parameters**

RF models can evaluate the importance of features (input parameters) by calculating the mean decrease impurity (MDI) for each feature among all the trees in the forest. The MDIs and corresponding standard deviations of each parameter at different supersaturations are shown in Fig. 5. Importance of the nine input parameters varies with supersaturations. For 0.07% and 0.10%, $\kappa_{\alpha355}$ and $\kappa_{\beta1064}$ are the two most important parameters, showing huge impact of hygroscopicity on the relationship

between CCN and optical properties. For 0.20%, $\mathring{a}_{\alpha355\&532}$ becomes much more important. Among the nine input parameters, $\kappa_\xi$ are denoted as hygroscopicity-related parameters, and $\mathring{a}_\xi$ are denoted as size-related parameters. Particularly, $s_a$ can be regarded as both size- and hygroscopicity-related parameter. As is shown in Fig. 5, hygroscopicity-related parameters, especially $\kappa_{\alpha355}$, $\kappa_{\beta1064}$, and $s_{a532}$, play crucial roles in retrieving CCN. Size-related parameters have already been proved to be vital in retrieving CCN, however, humidogram parameters $\kappa_\xi$ have not been implemented in previous methods. CCN

concentrations retrieved with and without $\kappa_\xi$ are compared to show the importance of $\kappa_\xi$. When retrieving CCN without $\kappa_\xi$, the RF model is also trained with datasets from campaign C1-C4, but the input data only contains Ångström exponents and lidar ratios. The retrieved CCN concentrations are all compared with datasets from campaign C5, and the results are listed in Table 5. $R^2$ of retrieved CCN decreases from 0.991 to 0.887 for supersaturations of 0.07%, from 0.992 to 0.857 for 0.10%, and from 0.973 to 0.785 for 0.20%. Retrieval errors also increase overwhelmingly, and there are significant positive systematic

biases. Parameters which are derived from backscatter and extinction enhancements, $\kappa_\xi$, are indispensable parameters in CCN retrieval.

**4.3 Impact of systematic and random error on CCN retrieval**

Figure 6 shows the relative errors of CCN retrieved with systematic errors in backscatter and extinction. Errors of retrieved CCN increase as errors of backscatter and extinction increase, and higher supersaturations are more affected by errors of optical

parameters. Errors in extinction coefficients at 355 nm ($\alpha_{355}$) influence the retrieval results most. In average, a positive relative error of 20% in $\alpha_{355}$ will cause about 20% overestimate in CCN number concentrations for supersaturation of 0.07%, about





40% overestimate for 0.10%, and about 60% overestimate for 0.20%. A negative error of 20% in $\alpha_{355}$ will underestimate CCN concentrations, and the degree of impact is slightly smaller than positive error. Errors in extinction coefficient at 532 nm ($\alpha_{532}$) and at 355nm have opposite effect on retrieval error. Bigger $\alpha_{355}$ means more small particles and higher number concentrations, and bigger $\alpha_{532}$ means more large particles. Errors in $\alpha_{532}$ do not show significant impact at supersaturations of 0.07% and

0.10%, but an overwhelming effect is found at supersaturations of 0.20%. It is interesting to note that the errors in backscatter coefficients do not affect the results much. However, in practical applications of MWRLs, the errors in extinction are always much larger than the errors of backscatter. If the error of retrieved CCN concentrations needs to be limited to 20% at supersaturation of 0.20%, the errors of retrieved extinction coefficients should to be controlled within 5%.

The test result of systematic error in RH is shown in Fig. 7. When RH has a negative systematic error, CCN concentrations

are overestimated, and the extent of overestimation increases as the error increase. A negative error of 10% in RH will overestimate CCN at supersaturations at 0.20% by about 60% in average, and the standard deviation is over 60%. Effects of positive errors in RH is much smaller than negative errors but more complicated. The standard deviations of retrieval relative error increase with RH error, and the extreme value of the mean retrieval error appears at the RH error of 5%. Underestimating RH will causes much more errors than overestimation. Great care should be paid to RH profiles if enhancements of backscatter

and extinction with RH are utilized.

The relative error of retrieved CCN with random errors are presented in Table 6. The mean values of relative error is -2.8%, -1.3%, and 1.3% for CCN at supersaturations of 0.07%, 0.10%, and 0.20%, respectively, and the corresponding standard deviations are 29.7%, 31.5%, and 42.9%. The impact of random errors on the nine input parameters is also evaluated and is shown in Fig. 8. Random errors (10% for backscatter and extinction, and 5% RH) underestimate $\kappa_\xi$ by 30%-35% in average,

and the standard deviations are about 40% or more. $s_{a355}$, $s_{a532}$, and $\mathring{a}_{\beta532\&1064}$ are overestimated by 5%-10%.

## 5 Summary

CCN number concentration at cloud base is a crucial and scarce parameter to constrain the relationship between aerosols and clouds. A new method to retrieve CCN number concentrations using backscatter and extinction coefficients from MWRL measurements is proposed. Enhancements of backscatter and extinction coefficients with RH are implemented to derive dry

backscatter/extinction $\xi_{\text{dry}}$ and humidogram parameter $\kappa_\xi$. The ratio of CCN number concentration to dry backscatter or extinction coefficient $\text{AR}_\xi$, which is estimated by $\kappa_\xi$, Ångström exponents, and lidar ratios, is introduced to retrieve CCN number concentrations.

The method is established and verified by theoretical simulations using Mie theory and $\kappa$-Köhler theory with in situ measured particle size distributions, mixing states, and chemical compositions. The values of $\text{AR}_\xi$ are found to have large variations due

to different size distributions and hygroscopicity. Theoretical analyses show that optical properties provided by current $3\beta+2\alpha$ MWRL systems basically contains size distribution and hygroscopicity information of particles with diameters larger than 100





nm, which only fits the critical diameters for supersaturations lower than 0.20%. Accordingly, CCN number concentrations at supersaturations of 0.07%, 0.10%, and 0.20% are retrieved. The performance of the new method is evaluated with error-free data, and CCN number concentrations at all three supersaturations are in good agreements with theoretical calculated values. Sensitivity tests are carried out to show the influence of systematic and random errors of lidar-derived optical properties and

auxiliary RH profiles on CCN retrieval. Systematic errors in extinction coefficients and RH are found to have large impact on error in retrieved CCN. Parameters fitted from backscatter and extinction enhancements (i.e. $\xi_{dry}$ and $\kappa_\xi$) is significantly influenced by RH. The uncertainty of RH profiles derived by remote sensing techniques is a major problem in CCN retrieval. Optical properties near cloud base from lidar measurements always influenced by high RH. Thus, transforming backscatter and extinction coefficients at ambient RH to dry conditions is a must for CCN retrieval, and accurate RH profiles are highly

demanded.

The importance of humidogram parameters $\kappa_\xi$ is demonstrated by comparing the error of CCN concentration retrieved with and without $\kappa_\xi$. Neglecting hygroscopicity information contained in backscatter and extinction enhancements will bring huge errors to CCN retrieval by lidars. The performance of two parameterization schemes for backscatter and extinction humidograms are evaluated. The $\kappa$-equation shows better performance on inferring dry backscatter and extinction than $\gamma$-

equation. The $\kappa$-equation, therefore, is recommended to describe the hygroscopic behaviors of the backscatter and extinction coefficients from lidar measurements. The fitted hygroscopic parameter are found to be sensitive to fitting RH range when the RH range is limited and relatively high (between 60%-90%). This is an extreme essential problem for current research for aerosol hygroscopicity with lidar measurements. Great care should be paid to the RH range when evaluating the hygroscopic growth of the lidar-related optical properties.

It should be note that the theoretical analyses in this paper are based on datasets of continental aerosols, and the implement of Mie theory also limits the scope of the results. The results can be applied in the North China Plain but are not fit for sea salts and mineral dust. Studies with datasets of other aerosol types should be carried out in the future. Although the applicability of this new method is limited by large uncertainties in RH profiles, comparison between real measured MWRL data and airborne in situ measurement should also be conducted.

This work furthers our understanding of the relationship between CCN and aerosol optical properties and providing an optional way to retrieve CCN number concentration profiles from lidar measurements. The newly proposed method has potential to provide long-term CCN at cloud base for aerosol-cloud-interaction studies.

*Author contribution.* C. Zhao and C. Li determined the main goal of this study. W. Tan and G. Zhao designed the methods. W.

Tan carried them out and prepared the manuscript with contributions from all co-authors.

*Competing interests.* The authors declare that they have no conflict of interest.





*Acknowledgements.* The study is supported by the National Key R&D Program of China (2016YFC0202000: Task 4) and the National Natural Science Foundation of China (41375008, 41590872, 9154400001, 41527807).

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



**Table 1.** Locations, time periods, parameters, and instruments of five field campaigns

| **Location** | Wuqing | Wuqing | Xianghe | Xianghe | Wangdu |
|---|---|---|---|---|---|
| **Campaign name** | C1 | C2 | C3 | C4 | C5 |
| **Time period** | 7 March to 4 April, 2009 | 12 July to 14 August, 2009 | 22 July to 30 August, 2012 | 9 July to 30 August, 2013 | 4 June to 14 July, 2014 |
| **PNSD** | TSMPS+APS | TSMPS+APS | SMPS+APS | TSMPS+APS | TSMPS+APS |
| $m_{BC}$ | MAAP | | | | |
| **HBF** | TSI 3563 nephelometer | | | | |
| **Size-resolved chemical composition** | – | Substrates sampled by BLPI | – | | |

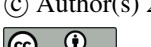



**Table 2.** Slopes of linear regressions, determination coefficients ($R^2$), and relative errors (RE) between Mie model simulated particle dry backscatter/extinction coefficients and those inferred from humidogram functions. 404575 pairs of data is used. The RE are given in the form of mean value ± one standard deviation (std).

| RH (%) | $\xi$ | $\gamma$-equation | | | $\kappa$-equation | | |
|---|---|---|---|---|---|---|---|
| | | slope | $R^2$ | RE(%) | slope | $R^2$ | RE(%) |
| 60-90 | $\alpha_{355,dry}$ | 0.850 | 0.998 | -16.2 ± 2.1 | 1.045 | 0.998 | 3.4 ± 2.4 |
| | $\alpha_{532,dry}$ | 0.820 | 0.998 | -19.2 ± 2.0 | 1.017 | 0.999 | 0.5 ± 1.8 |
| | $\beta_{355,dry}$ | 0.784 | 0.960 | -20.8 ± 7.2 | 0.817 | 0.971 | -9.6 ± 7.5 |
| | $\beta_{532,dry}$ | 0.812 | 0.972 | -22.7 ± 7.6 | 0.874 | 0.988 | -11.7 ± 5.6 |
| | $\beta_{1064,dry}$ | 0.878 | 0.986 | -12.9 ± 5.7 | 0.935 | 0.994 | -5.4 ± 4.4 |
| 60-70 | $\alpha_{355,dry}$ | 0.913 | 1.000 | -9.2 ± 1.1 | 1.016 | 1.000 | 1.1 ± 0.9 |
| | $\alpha_{532,dry}$ | 0.900 | 0.999 | -10.4 ± 1.3 | 1.005 | 1.000 | 0.0 ± 0.7 |
| | $\beta_{355,dry}$ | 0.939 | 0.989 | -9.1 ± 6.0 | 0.906 | 0.991 | -5.6 ± 4.9 |
| | $\beta_{532,dry}$ | 0.939 | 0.990 | -9.9 ± 5.6 | 0.939 | 0.996 | -6.4 ± 3.9 |
| | $\beta_{1064,dry}$ | 0.966 | 0.997 | -3.9 ± 2.9 | 0.974 | 0.999 | -1.9 ± 2.0 |
| 70-80 | $\alpha_{355,dry}$ | 0.852 | 0.999 | -15.8 ± 1.9 | 1.037 | 0.999 | 2.7 ± 2.1 |
| | $\alpha_{532,dry}$ | 0.827 | 0.998 | -18.3 ± 1.9 | 1.012 | 0.999 | 0.3 ± 1.5 |
| | $\beta_{355,dry}$ | 0.799 | 0.950 | -20.5 ± 8.9 | 0.818 | 0.968 | -10.5 ± 8.1 |
| | $\beta_{532,dry}$ | 0.833 | 0.966 | -21.4 ± 9.0 | 0.880 | 0.986 | -11.7 ± 6.6 |
| | $\beta_{1064,dry}$ | 0.898 | 0.987 | -10.8 ± 5.7 | 0.942 | 0.995 | -4.6 ± 4.1 |
| 80-90 | $\alpha_{355,dry}$ | 0.756 | 0.922 | -26.5 ± 3.8 | 1.110 | 0.991 | 8.5 ± 5.5 |
| | $\alpha_{532,dry}$ | 0.702 | 0.994 | -31.9 ± 3.1 | 1.047 | 0.995 | 1.9 ± 4.2 |
| | $\beta_{355,dry}$ | 0.547 | 0.848 | -37.0 ± 11.1 | 0.695 | 0.892 | -13.4 ± 14.1 |
| | $\beta_{532,dry}$ | 0.593 | 0.925 | -42.1 ± 8.7 | 0.775 | 0.961 | -19.2 ± 8.7 |
| | $\beta_{1064,dry}$ | 0.702 | 0.934 | -30.4 ± 10.3 | 0.867 | 0.971 | -11.5 ± 8.8 |



**Table 3.** Slopes of linear regressions and determination coefficients ($R^2$) between $\gamma_\xi$ or $\kappa_\xi$ fitted from RH range 60%-90% and those fitted from limited RH ranges (60%-70%, 70%-80%, and 80%-90%).

| RH (%) | $\xi$ | $\gamma_\xi$ slope | $\gamma_\xi$ $R^2$ | $\kappa_\xi$ slope | $\kappa_\xi$ $R^2$ |
|---|---|---|---|---|---|
| | $\alpha_{355}$ | 0.992 | 0.958 | 1.113 | 0.955 |
| | $\alpha_{532}$ | 0.969 | 0.978 | 1.007 | 0.977 |
| 60-70 | $\beta_{355}$ | 1.019 | 0.814 | 1.213 | 0.819 |
| | $\beta_{532}$ | 0.790 | 0.797 | 0.891 | 0.799 |
| | $\beta_{1064}$ | 0.806 | 0.834 | 1.011 | 0.812 |
| | $\alpha_{355}$ | 1.021 | 0.996 | 1.045 | 0.995 |
| | $\alpha_{532}$ | 1.015 | 0.997 | 1.014 | 0.997 |
| 70-80 | $\beta_{355}$ | 1.115 | 0.968 | 1.195 | 0.958 |
| | $\beta_{532}$ | 1.078 | 0.973 | 1.128 | 0.969 |
| | $\beta_{1064}$ | 0.999 | 0.979 | 1.034 | 0.972 |
| | $\alpha_{355}$ | 0.941 | 0.939 | 0.847 | 0.934 |
| | $\alpha_{532}$ | 0.957 | 0.969 | 0.969 | 0.967 |
| 80-90 | $\beta_{355}$ | 0.741 | 0.679 | 0.684 | 0.626 |
| | $\beta_{532}$ | 0.970 | 0.851 | 1.002 | 0.827 |
| | $\beta_{1064}$ | 1.090 | 0.816 | 1.036 | 0.818 |





**Table 4.** Lidar derived parameters for predicting optical-related CCN activation ratio $AR_\xi$

| Parameter | Description |
|---|---|
| $\kappa_{\alpha355}$ | Fitted parameter of extinction humidogram at 355 nm in $\kappa$-equation form |
| $\kappa_{\alpha532}$ | Fitted parameter of extinction humidogram at 532 nm in $\kappa$-equation form |
| $\kappa_{\beta355}$ | Fitted parameter of backscatter humidogram at 355 nm in $\kappa$-equation form |
| $\kappa_{\beta532}$ | Fitted parameter of backscatter humidogram at 532 nm in $\kappa$-equation form |
| $\kappa_{\beta1064}$ | Fitted parameter of backscatter humidogram at 1064 nm in $\kappa$-equation form |
| $s_{a355}$ | Particle dry lidar extinction-to-backscatter ratio at 355 nm |
| $s_{a532}$ | Particle dry lidar extinction-to-backscatter ratio at 532 nm |
| $\mathring{a}_{\alpha355\&532}$ | Ångström exponent of particle dry extinction coefficients between 355 and 532 nm |
| $\mathring{a}_{\beta532\&1064}$ | Ångström exponent of particle dry backscatter coefficients between 532 and 1064 nm |





**Table 5.** Slopes of linear regressions, determination coefficients ($R^2$), and relative errors (RE) between theoretical calculated CCN number concentrations and CCN number concentrations retrieved with/without $\kappa_\xi$ as input parameter. The relative error are given in the form of mean value $\pm$ one standard deviation (std).

| Supersaturation Ratio | With $\kappa_\xi$ | | | Without $\kappa_\xi$ | | |
|---|---|---|---|---|---|---|
| | slope | $R^2$ | RE(%) | slope | $R^2$ | RE(%) |
| **0.07%** | 0.991 | 0.991 | -0.8 ± 6.0 | 0.877 | 0.866 | 4.6 ± 26.1 |
| **0.10%** | 0.992 | 0.989 | 0.1 ± 6.3 | 0.857 | 0.837 | 5.9 ± 26.7 |
| **0.20%** | 1.005 | 0.973 | 3.9 ± 9.0 | 0.860 | 0.785 | 11.9 ± 28.1 |




**Table 6.** Mean and one standard deviation (std) values of relative errors in retrieved CCN number concentrations at different supersaturations with error-free and random error (10% for backscatter/extinction and 5% for relative humidity) conditions.

| Supersaturation Ratio | Error-free (mean ± std) | Random Error (mean ± std) |
|---|---|---|
| 0.07% | -0.8% ± 6.0% | -2.8% ± 29.7% |
| 0.10% | 0.1% ± 6.3% | -1.3% ± 31.5% |
| 0.20% | 3.9% ± 9.0% | 1.3% ± 42.9% |



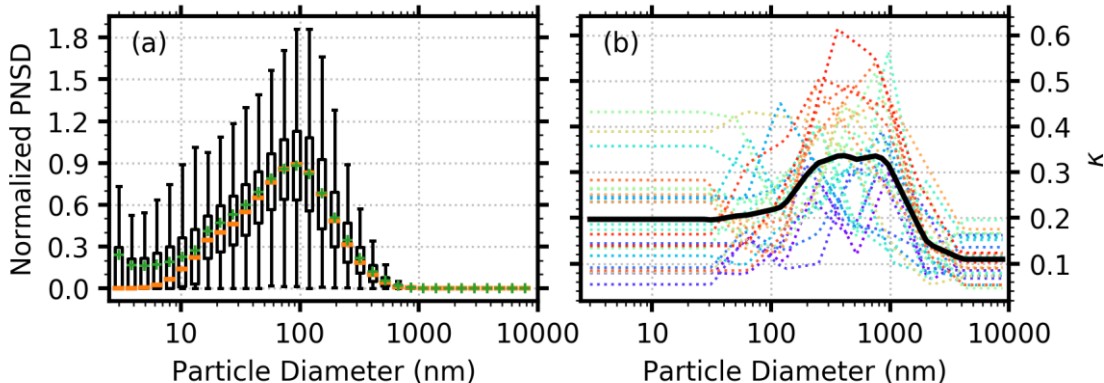

**Figure 1. (a)** Boxplot of particle number size distributions (PNSDs) in the datasets from five field campaigns. Each PNSD is normalized by its number concentration of total particles. Green markers "+" represent the mean value of each diameter. The boxes extend from the lower to upper quartile values, with orange lines at the median. The whiskers extend from the box to the minimum/maximum values or extend from the box by 1.5 times of interquartile range. The flyers are not shown in the plot.
**(b)** Twenty-five typical size-resolved $\kappa$ distributions. Each dotted line with color represents one size-resolved $\kappa$ distribution. The solid black line represents the mean value of the size-resolved $\kappa$ distributions.




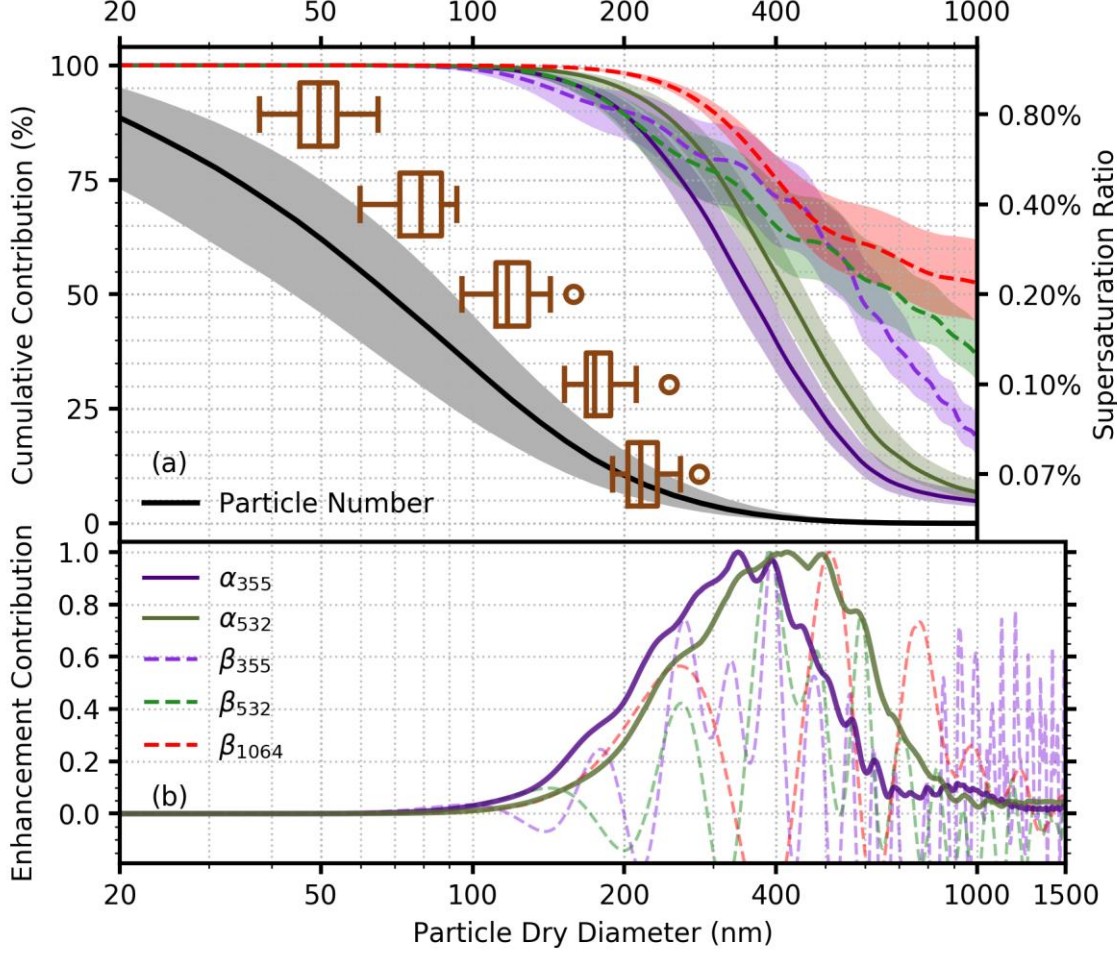

**Figure 2. (a)** Cumulative contributions (accumulate from large particle size to small particle size) of particle number concentrations (measured), dry particle backscatter coefficients (simulated), and dry particle extinction coefficients (simulated). The solid and dashed lines represent the median values of five field campaigns, and the shadows cover from the lower to upper quartile values. The box plots in brown contain statistical information about critical diameter of each supersaturation condition (right y-axis). The boxes extend from the lower to upper quartile values, with lines at the median. The whiskers extend from the box to the minimum/maximum values or extend from the box by 1.5 times of interquartile range. The markers "o" are the flyers. **(b)** Normalized size-resolved enhancement contributions when relative humidity increase from 60% to 90%, which are theoretically calculated by the mean particle number size distribution, the mean black carbon mass concentration (4.717 μg m⁻³), the mean mass ratio of externally mixed black carbon (0.664%), and the mean size-resolved $\kappa$ distribution.





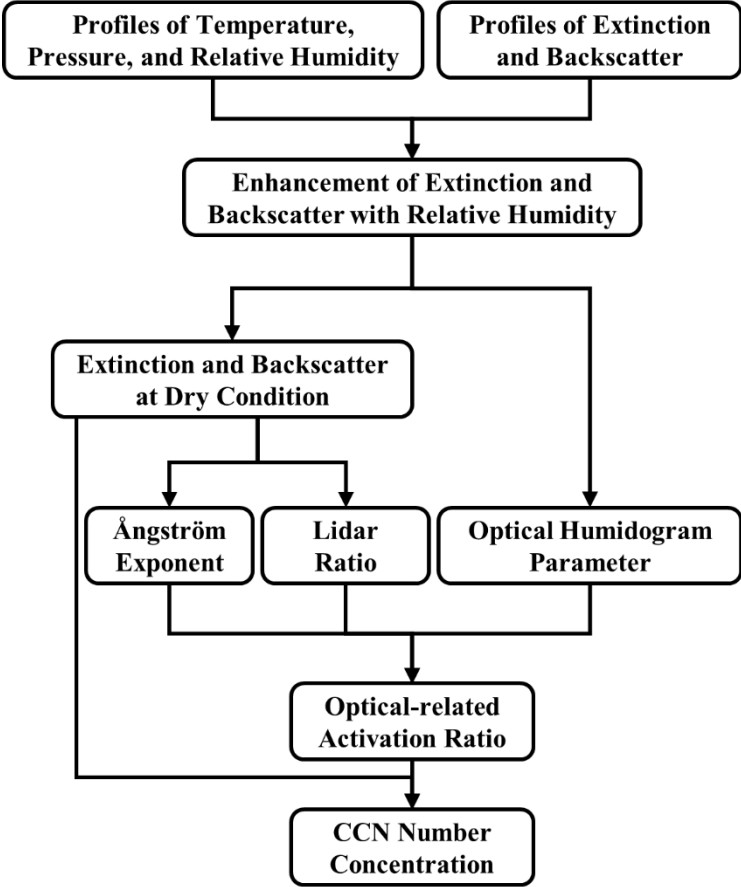

**Figure 3.** Schematic diagram of newly proposed method to retrieve cloud condensation nuclei number concentrations using multiwavelength Raman lidar.



**Figure 4.** Comparison of the theoretical calculated extinction-related CCN activation ratio at 532 nm and the model predicted extinction-related CCN activation ratios at 532 nm at supersaturations of **(a)** 0.07%, **(c)** 0.10%, and **(e)** 0.20%, and of the theoretical calculated CCN number concentrations and the retrieved CCN number concentrations at supersaturations of **(b)**





0.07%, **(d)** 0.10%, and **(f)** 0.20%. A total of 80575 pairs of data calculated from campaign C5 are used. The solid line is 1:1 line, and the dashed lines are 20% relative difference lines. Colors represent the relative density of the data points normalized by the maximum data density of each panel. The relative error showed in the figure is mean value ± one standard deviation.




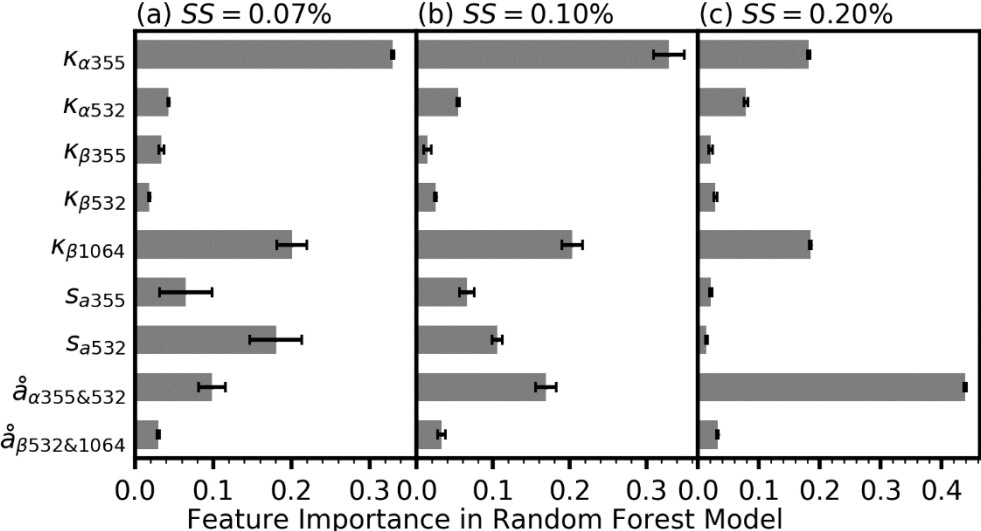

**Figure 5.** Importance of each feature (input parameter) output by the Random Forest model for predicting optical-related CCN activation ratios at supersaturations of **(a)** 0.07%, **(b)** 0.10%, and **(c)** 0.20%. The values of feature importance indicate the decrease in impurity for each feature. The length of bars represent the mean values among all trees and the error bars give the standard deviations.



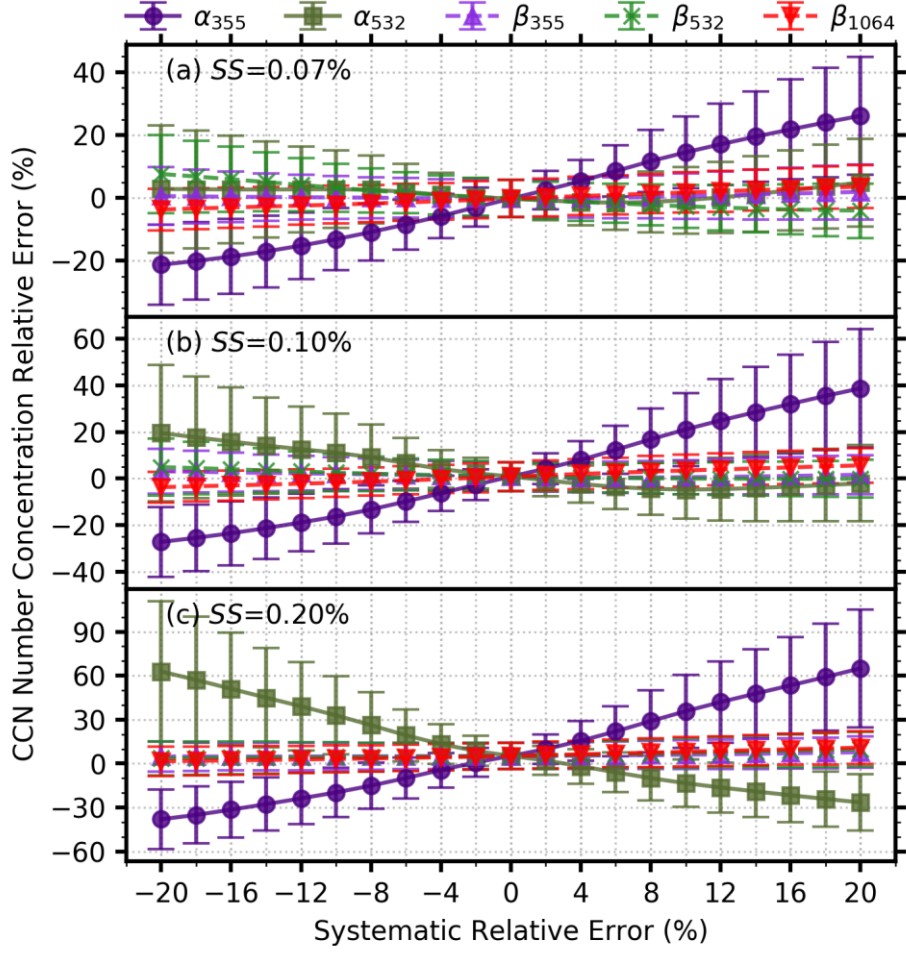

**Figure 6.** Relative errors in retrieved CCN number concentrations at supersaturations of **(a)** 0.07%, **(b)** 0.10%, and **(c)** 0.20% as a function of systematic errors in backscatter or extinction. The markers are the mean values, and the error bars denote the standard deviations.





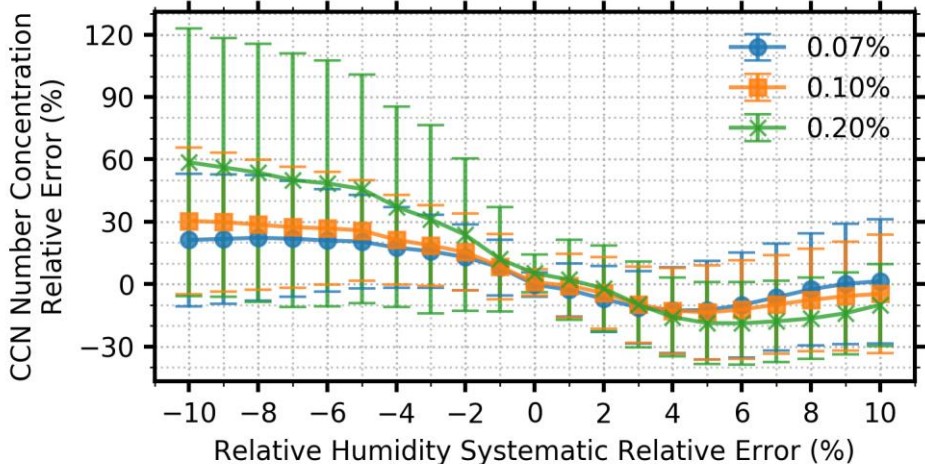

**Figure 7.** Relative errors in retrieved CCN number concentrations at supersaturations of 0.07%, 0.10%, and 0.20% as a function of systematic errors in relative humidity. The markers are the mean values, and the error bars denote the standard deviations.




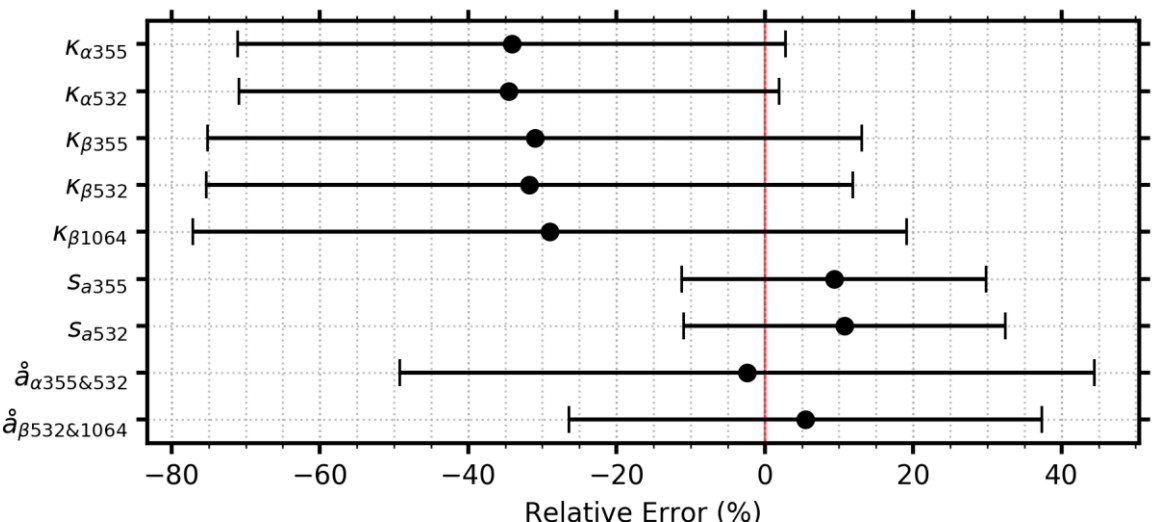

**Figure 8.** Relative errors in fitted and calculated parameters with 10% random errors for backscatter/extinction and 5% random error for relative humidity. The dots are the mean values, and the error bars denote the standard deviations.