# Peer review of "Method to retrieve cloud condensation nuclei number concentrations using lidar measurements"

_Atmospheric Measurement Techniques, 2019_

## Referee Comment (RC1) · Anonymous Referee #1 · 11 Mar 2019

**General comments**

The authors have developed a new approach to retrieve CCN number concentrations. It is my opinion that this topic is one of the most important for tackling the uncertainties surrounding climate radiative forcing. The method here is unique in that it utilizes some machine learning along with lidar and in-situ measurements to derive CCN number concentration. I found the work to be interesting and of high quality, but the methods and figure presentation need some revision and/or clarification. After my comments and concerns have been addressed, I feel the manuscript will be suitable for publication in AMT. Therefore, I recommend acceptance after some minor revision.

**Major comments**

[Figure]

One major point I would like to see addressed is on the performance of the humido-gram parameter estimates. The $N_{CCN}$ retrieval heavily relies on calculated dry optical parameters (dry angstrom exponent, dry lidar ratio) which are determined from the fitted dry extinction/backscatter, and the $\kappa$ parameter. I'd like to see a figure of example MWRL profiles that were used in this study with the model fit lines. A statistical summary plot would suffice if there is large scatter. The actual fits and profiles would be highly beneficial for me as the reader to visually assess the fit performance and also validate the layer selection that was mentioned in section 3.3.2.

Also in reference to the above comment, Table 3 is confusing to me. I'm not sure what information is gained by partitioning the humidogram curve in a way that ignores the high or the low end and then comparing the "partial" fit coefficient to the full range fit coefficient, since it is the entirety of the curve that describes the aerosol chemistry/size distribution properties. I recommend instead, that the authors show the parameterization statistics to the fitted data (e.g. RMSE or another metric) rather than what is shown now in Table 3.

I can appreciate investigating the performance of the parameterizations as a function of the RH range. But I don't think the breakdown here is important. The $\kappa$ parameterization has been well compared to the $\gamma$ parameterization as the authors note with the Brock et al. (2016) reference. Of more interest to me is the visual performance of these fits. I'd still like to see these in addition to the discussion and the Table entries that are already in the manuscript.

**Figure 1.** I think this figure could be constructed instead by normalizing by the maximum value at the peak diameter instead of total number. As it is constructed now, the range in the y-axis values makes this figure hard to interpret. If normalizing by the maximum value at the peak diameter (so that each distribution peaks at 1 rather than something less than 1) doesn't result in much change, you could also consider a

time-series with diameter on the y-axis and colors representing normalized PNSD.

**Minor comments**

**Table 4 and Text** The 9 parameters that are selected to determine $AR_\xi$ are declared to have no explicit expressions between them and have highly non-linear relationships (pg. 8 line 23-24). It's not clear that the normalized extinction at 532 nm and at 355 nm, for example, would have different enough $\kappa$ fit parameters to yield information for $AR_\xi$. Could you comment on this or possibly add a supplementary figure that would give support to the statement that there is no explicit expression between the 9 selected parameters? Worded another way, what information content can be gained from a spectrally dependent hygroscopicity fit parameter?

**Training set** I think the exercise would be more convincing if you divided your entire dataset into a training and test class where, for example, 70% of the data is randomly chosen for training and the other 30% is used for the test performance.

The Mie model results are calculated for the entire range of 25 kappa size resolved distributions but how is the final result selected for comparison to the MWRL retrieval method? Could you more clearly state this somewhere appropriate? Could you also, before section 2.2, explain the significance of a "size-resolved" kappa distribution? Kappa is thought to be size-independent for particles of certain chemical composition. It might be important to include a sentence or two explaining that the size-resolved kappa distribution is for particles of changing chemical composition with the Liu et al. (2014) reference.

**Technical corrections**

**Table 2.** Can you clarify in the caption if the Mie model simulated dry parameters are from the measured PNSDs?

**Pg. 1 Line 18** change "datasets" to "dataset"

**Pg. 1 Line 25** change "a huge" to "an"

**Pg. 2 Line 8** change "always" to "can"

**Pg. 2 Line 10** add the word in brackets: "in [the] natural"

**Pg. 2 Line 13** change "showing" to "suggesting"

**Pg. 2 Line 18** add characters in brackets: "Existing approach[es]"

**Pg. 2 Line 27** add the word in brackets: "...humidified in [the] ambient..."

**Pg. 3 Line 5** add the word in brackets: "...hints [at] the ability..."

**Pg. 3 Line 10** remove sentence beginning "Enhancements of ..."

**Pg. 3 Line 13** add an "s" to scheme and humidogram

**Pg. 7 Line 17** can you provide a reference for the backscatter angstrom exponent relationship?

**Pg. 7 Line 28-29** Rewrite the sentence beginning "Particle type information..." as follows: "The lidar ratio can provide information on particle type and also serve as a proxy for particle hygroscopicity. Therefore, the lidar ratio of dry particles could be a reliable parameter to estimate AR$_\xi$." or something like this.

**Pg. 8 Line 1** change "huge" to "large".

**Pg. 8 Line 26** add characters in brackets: "...been a field that [has] develop[ed] rapidly..."

**Pg. 10 Line 8** add characters in brackets: "...lidars may not [be] sufficient enough..."

**Pg. 10 Line 14** change "huge" to "the"

**Pg. 10 Line 30** change "In average" to "On average"

**Pg. 11 Line 3** rewrite the sentence begging with "Bigger...." This needs to clearly state that smaller particles have larger angstrom exponents and bigger particles have smaller angstrom exponents (or more compactly, extinction angstrom exponents are inversely proportional to particle size). Do you mean here that $\alpha_{355:532} > \alpha_{532:1064}$ means smaller particles? I'm not sure if that's true since the relationship is complex (e.g. Schuster et al., 2006; doi:10.1029/2005JD006328)

**Pg. 12 Line 20** change to "It should be [noted]...."

---

## Referee Comment (RC2) · Anonymous Referee #2 · 26 Mar 2019

General

The paper deals with an important topic of atmospheric research: The retrieval of CCN from multiwavelength backscatter and extinction lidar observations. New aspects are integrated (e.g., using the increase in backscatter and extinction coefficient with relative humidity, humidogram approach). This is an excellent idea!

However, the paper has to my opinion almost no structure. Although one gets an idea how the method may work, a well-elaborated and well-structured description of the methodology is completely missing. A huge amount of information on microphysical and chemical particle particle properties, simulated related optical effects, dependence of optical effects from increasing humidity (growth factors), etc. and respective experimental data from field observations in the North China Plain (NCP) are just accumulated in different sections, but the reader must already be an expert in this field to get an idea how all this may fit together.

So, the manuscript is far from being acceptable. To be short: to my opinion it cannot be published in the present version and must be rejected. In the following, I will give more detail so that the authors may get an idea how to change the structure of the manuscript and then resubmit it.

A manuscript with clear goals, clear structure, an easy-to-follow methodology, plus uncertainty analysis, and also some convincing but simple case studies are required.

An important issue is also: Please do not consider NCP aerosol only. It would be nice to have a general methodology that can be applied to anthropogenic haze and smoke, to dust aerosol, and may be to aerosols dominated by marine sea salt. All the required kappa values and growths factors are available in the literature. You do not need to focus on NCP aerosol!

Details

P3, L8-9: . . . The new method to retrieve CCN . . . is proposed based on kappa Koehler theory. . . .. There are many examples of such 'isolated' phrases, not further explained. So, often the reader is confronted with a lot of information and then with the question: What do they want to say? How will that be used? Only experts know what the message behind probably is.

Section 2: As a motivation it is mentioned: Section 2 introduces the measured and simulated data sets. Ok. But why do you introduce these data, why do you need these data, what is the goal, do you need them as input? All this is not explained. Nothing is clear. As a reader one wants to know the motivation for every section, before the content of the section is presented.

And after reading of section 2 and all the subsequent sections it becomes more or less clear: This method obviously only holds for North China Plain (NCP) pollution aerosol.

But is that then a useful method, for all the lidar scientists around the globe?

As a reader I expect a well elaborated algorithm applicable to all relevant aerosol types: What about pure marine conditions, what a about desert dust scenarios, what about forest fire smoke in the free troposphere? How does the method work in these cases? All this is not considered in this paper. Only NCP aerosol.

The BC mixing state is introduced. ... size resolved chemical composition all come from campaign C2....... and after transforming the ambient wet aerodynamic diameters into dry volume-equivalent diameters, size resolved kappa distribution were derived from measured size resolved chemical composition. Twenty five typical size-resolved kappa distributions in the NCP are computed and later used..... in the simulations. Who can follow? Why do you present all this? The methodlogy is still not presented yet! The paper is to more than 50% just an acccumulation of information, like in a laboratory book .... Section 2.2.

P4, Eq3 is introduced, taken from another paper, D is introduced as particle/droplet radius (but it is obviously the droplet radius). The equation contains: RH = 1 + SS with SS in %, so what is the unit of RH?

P4, L29: Then Mie theory is used with all the parameters introduced before including the confusing 25 kappa distributions. It is impossible to check the simulations in detail. We are forced to read another paper (Zhao et al, 2017). So, as a reader I am totally confused by the accumulation of all the information from field campaigns, simulations, partly explained in another paper. All particles are spherical which may make sense for North China pollution close to cloud base but what about aerosol mixtures with a lot of mineral dust, practically 75% of the aerosol in China) probably contains dust, what do we do in these cases?

P5, In Section 3, among the discussion of another set of new aspects (useful or not) the authors 'jump' to a new topic: Size-resolved enhancement contributions of backscatter and extinction are calculated to discuss hygroscopicity sensitive size of optical enhancement factor measurement. Ok!?! ...strong oscillations are found in size enhancement contributions of backscatter coefficients. OK! But what is the message behind all this? Where is the methodological concept? Where is all the mentioned information used?

P6: Now equations describing light scattering enhancement factors are introduced, no word about aerosol types and related differences in the enhancement factors. So, here it became finally clear to me the authors only develop their method for North China Plain pollution aerosol.

In conclusion, the authors accumulate and accumulate information ... from field campaigns, from papers, from own simulations, but leave the reader alone... with the question: Why do you present all this? Where is the flow chart with all input parameters needed to compute CCN from backscatter and extinction coefficients at several wavelengths, including uncertainty bars? We are already at the end of page 6, and no methodological concept is presented yet. Section 5 (Summary) comes close (given on page 11).

Now the methodology section starts, Sect. 3.3.

P7 L6: The equation provides the basic relationship between lidar information (backscatter and extinction coefficient) and N-CCN.

P7 L4: you write backscatter/extinction coefficient, but I believe you want to write backscatter and extinction ...., and not to use the ratio (backscatter/extinction). All this is confusing!

P7, Eq.7 and Eq.8, again new parameters are introduced, new discussion and uncertainty sources, but no clear flow chart what to do with all this information in detail (step byr step).

P8, L4-6: Here the entire method is summarized within three lines! We need to study Figure 3 that shows a flow chart. This is a SKTECH! ... and helps to understand the

method. But a clear set of equations with all input parameters needed in the first step and all the output parameters, which are again input for the next step and so on, is missing. All this is needed for five optical properties (3 beta and 2 ext). All this is not presented. What about uncertainties in the retrieval? How can we get a convincing opinion on the potential (and especially the limits) if we have only Fig 3 and then the correlation plots in Figure 4. Figures 6 and Figure 7 are useful. But we need to see the overall concept (equations, including uncertainty computation approaches.) And we need it for other aerosol types (dust, haze, marine..), not only for NCP aerosol which is of course an important aerosol mixture.

---

## Author Comment (AC1) · 27 Apr 2019

Reply to Anonymous Referee #1

**General comments**
The authors have developed a new approach to retrieve CCN number concentrations. It is my
opinion that this topic is one of the most important for tackling the uncertainties surrounding climate
radiative forcing. The method here is unique in that it utilizes some machine learning along with
lidar and in-situ measurements to derive CCN number concentration. I found the work to be
interesting and of high quality, but the methods and figure presentation need some revision and/or
clarification. After my comments and concerns have been addressed, I feel the manuscript will be
suitable for publication in AMT. Therefore, I recommend acceptance after some minor revision.

**Response:**
Thanks for your encouraging comments. The manuscript has been revised according to your
suggestions. Please see the responses to the specific comments.

**Major comments**
One major point I would like to see addressed is on the performance of the humidogram parameter
estimates. The $N_{\text{CCN}}$ retrieval heavily relies on calculated dry optical parameters (dry angstrom
exponent, dry lidar ratio) which are determined from the fitted dry extinction/backscatter, and the $\kappa$
parameter. I'd like to see a figure of example MWRL profiles that were used in this study with the
model fit lines. A statistical summary plot would suffice if there is large scatter. The actual fits and
profiles would be highly beneficial for me as the reader to visually assess the fit performance and
also validate the layer selection that was mentioned in section 3.3.2.

**Response:**
Our paper presents the methodlogy based on theoretical simulation, and the methodlogy is not
applied to real cases. All the results in our paper are only based on in situ measurements and there
is no real measured lidar data. The MWRL backscatter and extinction data mentioned in the paper
is all simulated using in situ data and Mie model. In case of more misunderstandings, we have added
an introduction at the beginning of Sect. 2:
*'Since it is not easy to accumulate large datasets of simultaneous measurements of lidars and*
*aircrafts, ground-measured aerosol microphysical and chemical data are used to simulate lidar-*
*derived backscatter and extinction coefficients and corresponding CCN number concentrations.*
*The simulations are based on $\kappa$-Köhler theory and Mie theory. The required datasets include:*
*particle number size distribution (PNSD), black carbon (BC) mass concentrations ($m_{BC}$), mixing*
*state of BC containing particles, and size-resolved hygroscopicity. The simulation results are used*
*to establish and validate the new retrieval method.'*
Our method is based on many previous studies about aerosol hygroscopicity using lidar techniques
(Wulfmeyer and Feingold, 2000;Feingold and Morley, 2003;Pahlow et al., 2006;Fernández et al.,
2015;Rosati et al., 2016;Fernández et al., 2017;Haarig et al., 2017;Lv et al., 2017;Bedoya-Velásquez
et al., 2018). An example from Bedoya-Velásquez et al. (2018) is shown below. Figure 3 from
Bedoya-Velásquez et al. (2018) (Fig. R1) explains how to derive backscatter enhancement factors
using lidar-retrieved backscatter profiles and RH profiles with the selection criteria in Sect. 3.3.2.
The method in our paper mainly focus on the procedure start from the variations of backscatter and
extinction with RH. We assume that Mie model simulated dataset can represent actual lidar
measurements. Figure 7 from Bedoya-Velásquez et al. (2018) (Fig. R2) shows lidar observation and model simulation are in good agreements, especially for RH below 90%. Therefore, we think it is reasonable to use Mie model simulated backscatter and extinction at different RH in this study.

[Figure]

**Figure 3. (a, e)** Profiles of RH retrieved from RS (black line) and by the synergy RL + MWR (red line), **(b, f)** RH bias profiles (cyan line), **(c, g)** $\beta_{par}$ retrieved by using the Klett–Fernald algorithm and lidar ratio of 65 Sr (green line), and **(d, f)** $f_\beta$ (RH) calculated for RS (black dots) and by the synergy RL + MWR (red dots) and the corresponding Hänel parameterizations (solid lines), where the red line refers to the RL + MWR method (case I: $\gamma = 0.59 \pm 0.05$, case II: $\gamma = 0.95 \pm 0.02$) and the black line refers to the RS method (case I: $\gamma = 0.56 \pm 0.01$, case II: $\gamma = 0.99 \pm 0.01$). The top row corresponds to case I (22 July 2011, 20:30–21:00 UTC) and the bottom row to case II (22 July 2013, 20:00–20:30 UTC). Horizontal dashed lines indicate the altitude range analyzed for each case (1.3 to 2.3 km for case I and 1.3 to 2.7 km for case II). All these profiles were measured at the EARLINET IISTA-CEAMA station.

**Figure R1.** Example of deriving backscatter enhancement factors using lidar-retrieved backscatter profiles and RH profiles. (Figure 3 in Bedoya-Velásquez et al. (2018))

[Figure]

**Figure 7.** Humidograms calculated **(a)** at 532 nm and **(b)** at 355 nm, within the 1.5 to 2.4 km a.s.l. aerosol layer from the RL + MWR measurements and calculated using Mie theory and measured chemical composition and size distribution at 2.5 km a.s.l. $RH_{ref} = 78\%$ was used for both methods.

**Figure R2.** Comparison of lidar-derived humidogram curves and modelled humidogram curves.

(Figure 7 in Bedoya-Velásquez et al. (2018))

Here we use Mie model simulation to show the performance of the two humidogram
parameterization. Figure R3 gives an example of humidogram fitting. All the dots represent Mie
model simulations, and the dots in red (within RH range of 60-90%) are used to fit parameterization
lines. The blue line is the result of $\gamma$-equation, and the green line represents the result of $\kappa$-equation.
Both equations fit quite well for RH range of 60-90%. However, $\kappa$-equation has a better performance
on estimating optical properties at dry condition. The figure has been added to supplement file of
the paper.

[Figure]

**Figure R3.** Example of humidogram fitting using different functions. The example is calculated
with one set of PNSD, BC, $r_{ext}$, and size-resolved $\kappa$ distribution.

Figure R4 gives the performance on the estimation of dry backscatter and extinction (Same as Table
2). The figure only shows the results of RH range 60-90%. Dry optical properties fitted with $\kappa$-
equation agree better with Mie model simulations, especially for extinction. The figure has been
added to supplement file of the paper.

[Figure]

**Figure R4.** Comparison between Mie model calculated dry particle backscatter and extinction and those fitted from humidograms.

Also in reference to the above comment, Table 3 is confusing to me. I'm not sure what information is gained by partitioning the humidogram curve in a way that ignores the high or the low end and then comparing the "partial" fit coefficient to the full range fit coefficient, since it is the entirety of the curve that describes the aerosol chemistry/size distribution properties. I recommend instead, that the authors show the parameterization statistics to the fitted data (e.g. RMSE or another metric) rather than what is shown now in Table 3.

**Response:**

Yes, it is the whole curve that describes the aerosol properties. The motivation of showing Table 3 is based on practical situation of lidar observations. Unlike in situ measurements, hygroscopicity studies based on lidar measurements are facing a problem that we cannot manually control the RH we measure. The RH range that lidar observation can get is limited by the actual RH profile and the well-mixed assumption. Therefore, for every hygroscopic case of lidar, the RH ranges can be limited and very different. Humidogram parameters fitted from backscatter and extinction enhancements with RH (i.e. $\gamma_\xi$ or $\kappa_\xi$) are often used to represent the hygroscopicity of particles. However, if large difference in $\gamma_\xi$ or $\kappa_\xi$ for different RH range accrues for the same group of particles, then the humidogram parameters may not be comparable for different cases that have different RH ranges.

That is what we want to stress through the comparison in Table 3.

For our study, the values of $\kappa_\xi$ are important for retrieving CCN. The RH range is also important.

For example, the random forest model is now trained by $\kappa_\xi$ fitted from RH range of 60-90%, and the data collect from lidar measurements only contains RH within 80-90%. The results are presented in Fig. R5. Compared to the results in Fig. 4 in the paper, more uncertainties will arise if RH range is different between the training and test data.

[Figure]

**Figure R5.** Same as Fig. 4 in the paper. The training data is the same as Fig. 4 which is derived from RH range of 60-90%, but the test data is derived from RH range of 80-90%.

I can appreciate investigating the performance of the parameterizations as a function of the RH range. But I don't think the breakdown here is important. The $\kappa$ parameterization has been well compared to the $\gamma$ parameterization as the authors note with the Brock et al. (2016) reference. Of more interest to me is the visual performance of these fits. I'd still like to see these in addition to the discussion and the Table entries that are already in the manuscript.

**Response:**

As you addressed, the performance of the two parameterization has been well evaluated by Brock et al. (2016). Most of the comparisons focus on the performance on describing enhancement values at different RH, especially high RH, so we did not show many evaluation results on this performance. You can see both parameterizations fit quite well for high RH ranges from the figure in Fig. R3 and Fig. R4. Compared to the accuracy at high RH values, the estimated dry optical properties are much more important to our method. Therefore, we paid more attention to the performance on estimating dry backscatter and extinction. Through the investigation of the performance as a function of RH range, we found that if the data has lower RH, the estimated dry backscatter and extinction will be closer to the simulated values (Table 2). As for the visual performance, we have put Fig. R3 and Fig. R4 to the supplement file.

**Figure 1.** I think this figure could be constructed instead by normalizing by the maximum value at the peak diameter instead of total number. As it is constructed now, the range in the y-axis values makes this figure hard to interpret. If normalizing by the maximum value at the peak diameter (so that each distribution peaks at 1 rather than something less than 1) doesn't result in much change, you could also consider a time-series with diameter on the y-axis and colors representing normalized PNSD.

**Response:**

Figure 1a is now reconstructed by normalizing the PNSD by the maximum value at the peak diameter (Figure R6). The time series of the normalized PNSD is shown in Fig. R7. The figure has been added to the Supplement.

[Figure]

**Figure R6.** Reconstructed Fig. 1.

[Figure]

**Figure R7.** Time series of the normalized PNSD.

**Minor comments**

**Table 4 and Text** The 9 parameters that are selected to determine $AR_\xi$ are declared to have no explicit expressions between them and have highly non-linear relationships (pg. 8 line 23-24). It's not clear that the normalized extinction at 532 nm and at 355 nm, for example, would have different enough $\kappa$ fit parameters to yield information for $AR_\xi$. Could you comment on this or possibly add a supplementary figure that would give support to the statement that there is no explicit expression between the 9 selected parameters? Worded another way, what information content can be gained from a spectrally dependent hygroscopicity fit parameter?

**Response:**

Thanks for your comment and suggestion. There should be some linear relationship between some parameters. Figure R8 shows the relationship between fitted humidogram parameter of extinction ($\kappa_{\alpha355}$) and other 8 parameters.

[Figure]

**Figure R8.** Relationship between $\kappa_{\alpha355}$ and other 8 parameters.

High correlation ($R^2$=0.880) is found between humidogram parameters fitted from extinction at 355
nm and 532 nm. $R^2$ between $\kappa_{\alpha355}$ and $\kappa_{\beta1064}$ is only 0.126 which quite fits the results from Fig.
5 in the paper. The two parameters with low correlation are both important to predict $AR_\xi$.
We also present Table R1 to show the relationship between every two of the nine parameters. The
table indicates that some parameters are linear correlated to some extent. Accordingly, we remove
the sentence you mention.

**Table R1.** Determine coefficients ($R^2$) between input parameters. $R^2$ larger than 0.5 are marked in
red.

| $R^2$ | $\kappa_{\alpha355}$ | $\kappa_{\alpha532}$ | $\kappa_{\beta355}$ | $\kappa_{\beta532}$ | $\kappa_{\beta1064}$ | $s_{a355}$ | $s_{a532}$ | $\mathring{a}_{\alpha355\&532}$ |
|---|---|---|---|---|---|---|---|---|
| $\kappa_{\alpha355}$ | – | – | – | – | – | – | – | – |
| $\kappa_{\alpha532}$ | 0.880 | – | – | – | – | – | – | – |
| $\kappa_{\beta355}$ | 0.411 | 0.321 | – | – | – | – | – | – |
| $\kappa_{\beta532}$ | 0.508 | 0.644 | 0.450 | – | – | – | – | – |
| $\kappa_{\beta1064}$ | 0.126 | 0.222 | 0.016 | 0.056 | – | – | – | – |
| $s_{a355}$ | 0.085 | 0.073 | 0.680 | 0.292 | 0.019 | – | – | – |
| $s_{a532}$ | 0.026 | 0.070 | 0.117 | 0.423 | 0.070 | 0.360 | – | – |
| $\mathring{a}_{\alpha355\&532}$ | 0.149 | 0.135 | 0.505 | 0.267 | 0.027 | 0.627 | 0.089 | – |
| $\mathring{a}_{\beta532\&1064}$ | 0.062 | 0.023 | 0.550 | 0.169 | 0.464 | 0.409 | 0.023 | 0.317 |

As for the information content of spectrally dependent $\kappa_\xi$, we have tried to explain it with Fig. 2b
in the paper. The enhancement contribution is influenced by both hygroscopicity and number
concentration of each size. Still, from the figure, we can see different $\kappa_\xi$ is sensitive to the
hygroscopicity of different size. We want to obtain size-dependent hygroscopicity information with
the use of spectrally dependent $\kappa_\xi$, because size-dependent hygroscopicity is important to estimate
CCN rather than a bulk hygroscopicity information, especially for different supersaturation
conditions. One humidogram may indicate the bulk hygroscopicity, but it is the hygroscopicity of
small particles that influences CCN number concentrations most. Spectrally dependent $\kappa_\xi$ can provide some information about the hygroscopicity of small particles. Also from Fig. 5 in the paper which show the relative importance of the input parameters for random forest model, humidogram parameters of extinction at 355 nm and backscatter at 1064 nm are rather important. Therefore, at least these two parameters are needed for the retrieval method. We have added the explanation above to the corresponding text.

**Training set** I think the exercise would be more convincing if you divided your entire dataset into a training and test class where, for example, 70% of the data is randomly chosen for training and the other 30% is used for the test performance.

**Response:**
It is true that many studies randomly select their training and test class. One thing I am concerning here is that our dataset is from continuously in situ measurements. Two observations of very close time are likely to be similar. If the dataset is divided randomly, the random forest model can possibly be trained and tested with two very similar dataset. We have tried random selection strategy and found that the test results are better than the strategy used now in the paper. Good agreements are found even in high supersaturations if the data is divided randomly for training and testing. Thus, we think the random selection strategy is not appropriate for our data. We train the model with data from some sites and test with data from other sites, in order to test the feasibility of the method at different locations. There are some papers used the same strategy as we do (Kuang et al., 2018;Zhao et al., 2018).

The Mie model results are calculated for the entire range of 25 kappa size resolved distributions but how is the final result selected for comparison to the MWRL retrieval method? Could you more clearly state this somewhere appropriate? Could you also, before section 2.2, explain the significance of a "size-resolved" kappa distribution? Kappa is thought to be size-independent for particles of certain chemical composition. It might be important to include a sentence or two explaining that the size-resolved kappa distribution is for particles of changing chemical composition with the Liu et al. (2014) reference.

**Response:**
Thanks for your suggestion.
Since no concurrent measured microphysical and chemical properties, we conducted the simulation with all 25 size-resolved $\kappa$ distributions for every PNSD and did not select them for application. We trained the random forest model with all the results of 25 size-resolved $\kappa$ distributions. Actually, there should be some relationship between particle dry optical properties and their hygroscopicity (e.g. black carbon influences both lidar ratio and hygroscopicity very much). More details can be discussed if more concurrent measured data is available in the future. In the paper, we applied all 25 distributions to every PNSD in order to cover various situations. Training the random forest model without selection, to some extent, is reasonable because the simulations contain all situations. We have added the following statement at the end of Sect. 2.2:
*'The new method and all the analyses in this paper are based on the Mie model simulated datasets, and all the simulations mentioned above are implemented.'*
And we also added a sentence at the end of Sect. 2.1 to explain the size-resolved $\kappa$ distribution:
*'The chemical compositions are found to be size dependent during the campaign C2, especially the mass fraction of organic matter (Liu et al., 2014).'*

**Technical corrections**

**Table 2.** Can you clarify in the caption if the Mie model simulated dry parameters are from the measured PNSDs?

**Response:**

Thanks for your suggestion. We have rewritten the caption:

*'Slopes of linear regressions, determination coefficients ($R^2$), and relative errors (RE) between Mie model simulated particle dry backscatter or extinction coefficients and those inferred from humidogram functions. 404575 pairs of the simulations from in situ dataset are used. The RE are given in the form of mean value ± one standard deviation (std).'*

**Pg. 1 Line 18** change "datasets" to "dataset"

**Pg. 1 Line 25** change "a huge" to "an"

**Pg. 2 Line 8** change "always" to "can"

**Pg. 2 Line 10** add the word in brackets: "in [the] natural"

**Pg. 2 Line 13** change "showing" to "suggesting"

**Pg. 2 Line 18** add characters in brackets: "Existing approach[es]"

**Pg. 2 Line 27** add the word in brackets: "...humidified in [the] ambient..."

**Pg. 3 Line 5** add the word in brackets: "...hints [at] the ability..."

**Pg. 3 Line 10** remove sentence beginning "Enhancements of ..."

**Pg. 3 Line 13** add an "s" to scheme and humidogram

**Response:**

Thanks! Corrections have been made according to your suggestions.

**Pg. 7 Line 17** can you provide a reference for the backscatter angstrom exponent relationship?

**Response:**

The following reference (Komppula et al. 2012) has been added to the corresponding text:

*'Komppula, M., Mielonen, T., Arola, A., Korhonen, K., Lihavainen, H., Hyvärinen, A. P., Baars, H., Engelmann, R., Althausen, D., Ansmann, A., Müller, D., Panwar, T. S., Hooda, R. K., Sharma, V. P., Kerminen, V. M., Lehtinen, K. E. J., and Viisanen, Y.: Technical Note: One year of Raman-lidar measurements in Gual Pahari EUCAARI site close to New Delhi in India – Seasonal characteristics of the aerosol vertical structure, Atmos. Chem. Phys., 12, 4513-4524, 10.5194/acp-12-4513-2012, 2012.'*

**Pg. 7 Line 28-29** Rewrite the sentence beginning "Particle type information..." as follows: "The lidar ratio can provide information on particle type and also serve as a proxy for particle hygroscopicity. Therefore, the lidar ratio of dry particles could be a reliable parameter to estimate $AR_\xi$." or something like this.

**Pg. 8 Line 1** change "huge" to "large".

**Pg. 8 Line 26** add characters in brackets: "...been a field that [has] develop[ed] rapidly..."

**Pg. 10 Line 8** add characters in brackets: "...lidars may not [be] sufficient enough..." Pg. 10 Line 14 change "huge" to "the"

**Pg. 10 Line 30** change "In average" to "On average"

**Response:**

Thanks! Corrections have been made according to your suggestions.

**Pg. 11 Line 3** rewrite the sentence begging with "Bigger...." This needs to clearly state that smaller particles have larger angstrom exponents and bigger particles have smaller angstrom exponents (or more compactly, extinction angstrom exponents are inversely proportional to particle size). Do you mean here that $\alpha_{355:532} > \alpha_{532:1064}$ means smaller particles? I'm not sure if that's true since the relationship is complex (e.g. Schuster et al., 2006; doi:10.1029/2005JD006328)

**Response:**

Thanks for your suggestion. The symbol '$\alpha_{355}$' here represents extinction coefficient at 355 nm but not Ångström exponent. What I mean here is just that if $\alpha_{355}$ is overestimated and backscatter and extinction at other wavelengths remain unchanged, the corresponding Ångström exponent will become bigger. The sentence has been removed since the result here is obvious and needs no more detailed discussion.

**Pg. 12 Line 20** change to "It should be [noted]...."

**Response:**

Thanks! Correction has been made.

[revised manuscript text omitted]

**S3 Theoretical simulations of CCN, lidar backscatter, and extinction**

**S3.1 Aerosol model assumptions**

We assume the aerosol particles act as follows:

(1) Aerosol particles are spherical, which means the simulation results are not appropriate for mineral dust.

(2) Particles are partially externally mixed and partially core-shell mixed. Only two kinds of aerosols are contained: pure black carbon (BC) and BC coated by non-light-absorbing components. Note that if $r_{ext} = 1$, there exists pure BC and pure non-light-absorbing particles. $r_{ext}$ is defined with Eq. (2) in the paper.

(3) The shape of BC mass size distribution (BCMSD) remains unchanged and the amount is related to the total BC mass concentration ($m_{BC}$). The fixed distribution comes from the average BCMSD obtained from Berner impactor measurements (Ma et al., 2012) and is shown in Fig. S3.

[Figure]

**Figure S3~2~.** Normalized size distribution of black carbon (BC) mass/volume concentration. The distribution is the average black carbon mass concentration obtained from Berner impactor measurements (Ma et al., 2012) and is normalized by the maximum value of the distribution.

(4) $r_{ext}$ is uniform among different particle sizes. Accordingly, number concentrations of externally mixed BC ($N_{ext}$) can easily be calculated from BCMSD and $r_{ext}$:

$$N_{ext}(D) = \frac{r_{ext} \cdot m_{BC}(D)}{\frac{\pi}{6} D^3 \cdot \rho_{BC}},$$ (S1)

where $D$ is diameter of the particle, $\rho_{BC}$ is the density of BC. In this study, $\rho_{BC}$ is assumed to be 1.5 g/cm³, which is also the density when retrieving $r_{ext}$.

(5) For each particle size, the diameters of BC cores ($D_{core}$) are the same and can be derived using the following equation:

$$D_{core}(D) = \sqrt[3]{\frac{6(1-r_{ext}) \cdot m_{BC}(D)}{\pi \rho_{BC} \cdot [N(D) - N_{ext}(D)]}},$$ (S2)

where $N(D)$ represents particle number size distribution (PNSD).

(6) The size-resolved $\kappa$ distributions represent the bulk hygroscopicity of core-shell mixed particles, and externally mixed BC particles do not take up water.

All these assumptions above are strong and, to some extent, inconsistent with the reality, but are certified to be reasonable for calculating aerosol optical properties. Plenty of works on aerosol optical closure studies (Ma et al., 2011;Ma et al., 2012) and aerosol optical simulations (Kuang et al., 2017;Kuang et al., 2018;Zhao et al., 2018) have been carried out with these aerosol model assumptions. In particular, Zhao et al. (2017) use the aerosol model to simulate lidar backscatter and extinction under different relative humidity (RH) conditions.

**S3.2 Calculations of CCN number concentrations using $\kappa$-Köhler theroy**

According to $\kappa$-Köhler theory, CCN number concentrations at a specific supersaturation level can be calculated by PNSD and size-resolved $\kappa$ distribution. Based on Eq. (3), the critical supersaturation ratio required to activate a particle is decided by corresponding $\kappa$ and $D_{\text{dry}}$. In other word, we can get a critical activation dry diameter $D_c$ with a given supersaturation ratio and size-resolved $\kappa$ distribution. Then CCN number concentration $N_{\text{CCN}}(SS)$ thereby can be calculated with Eq. (S3):

$$N_{\text{CCN}}(SS) = \int_{D_c(SS)}^{D_{\max}}[N(D) - N_{\text{ext}}(D)]\,dD , \tag{S3}$$

where $D_{\max}$ correspond to the upper bounds of the measured PNSD. Note that we regard externally mixed pure BC as non-hygroscopic particles, so $r_{\text{ext}}$ and $m_{\text{BC}}$ should also be involved to calculate $N_{\text{ext}}$, which needs to be subtracted. Otherwise, $N_{\text{CCN}}$ will be overestimated.

**S3.3 Calculations of particle backscatter and extinction coefficients at different RH using $\kappa$-Köhler theroy and Mie theory**

We use a modified BHMIE Fortran code and a modified BHCOAT Fortran code to calculate optical properties of homogeneous spherical particles and coated spherical particles, respectively. For a homogeneous spherical particle (i.e. externally mixed BC in this study), BHMIE can calculate particle scattering and extinction efficiency ($Q_{\text{sca}}$ and $Q_{\text{ext}}$) and scattering phase function with given light wavelength, particle diameter, and complex refractive index. For a coated spherical particle (i.e. core-shell mixed particle), diameters and complex refractive indices of both core and shell are needed in BHCOAT. The particle backscatter and extinction coefficients we need are derived from $Q_{\text{sca}}$, $Q_{\text{ext}}$, and scattering phase function at 180° $P(\pi)$:

$$\alpha = \sum_i \left[ \int_{D_{\min}}^{D_{\max}} \frac{1}{4} D^2 Q_{\text{ext}}(D,i)N(D,i)\,dD \right], \tag{S4}$$

$$\beta = \sum_i \left[ \int_{D_{\min}}^{D_{\max}} \frac{1}{16} D^2 Q_{\text{sca}}(D,i)P(\pi,D,i)N(D,i)\,dD \right], \tag{S5}$$

where $D_{\min}$ and $D_{\max}$ correspond respectively to the lower and upper bounds of the measured PNSD, and the index $i$ indicates the mixing state of particles, i.e. external or core-shell in this paper. Polarization of lidar emitted laser is neglected.

Complex refractive indices are essential for Mie scattering calculation. Aerosol complex refractive indices are related to chemical components, morphology, and wavelengths of light (Cotterell et al., 2017). Both real part and imaginary part of refractive indices vary a lot in real ambient environment (Shettle and Fenn, 1979). Wavelength dependency of refractive indices at wavelengths of 355 nm, 532 nm, and 1064 nm are not significant except for brown carbon (Shettle and Fenn, 1979; Bond et al., 2013). Neglecting the effect of brown carbon in this study, we simply assume that complex refractive indices of corresponding components do not change with wavelengths. The refractive indices of BC and non-light-absorbing component (shell) are set to be $1.8+0.54i$ (Ma et al., 2012) and $1.53+10^{-7}i$ (Wex et al., 2002), respectively.

Concerning aerosol hygroscopicity for core-shell mixed particles, diameters of BC cores $D_{core}$ remain unchanged, and particle diameters $D$ at different RH can be calculated with Eq. (3). The refractive index of the swelling shell ($\widetilde{m}_{shell}$) is calculated following the volume mixing law (Hanel, 1968):

$$\widetilde{m}_{shell} = f_{solute} \cdot \widetilde{m}_{solute} + (1 - f_{solute}) \cdot \widetilde{m}_{water}, \quad (S6)$$

where $\widetilde{m}_{solute}$ is the refractive index of solute (i.e. $1.53+10^{-7}i$ in this study), $\widetilde{m}_{water}$ is the refractive index of pure water ($1.33+10^{-7}i$), and $f_{solute}$ is the solute volume fraction of the in solution (shell), which is determined by Eq. (S7):

$$f_{solute} = \frac{D_{dry}^3 - D_{core}^3}{D^3 - D_{core}^3}. \quad (S7)$$

[Figure]

**Figure S4.** Example of humidogram fitting using different functions. The example is calculated with one set of PNSD, BC, $r_{ext}$, and size-resolved $\kappa$ distribution. The dots represent Mie model simulations, and the dots in red (within RH range of 60-90%) are used to fit parameterization lines. The blue line is the result of $\gamma$-equation, and the green line represents the result of $\kappa$-equation.

[Figure]

**Figure S5.** Comparison between Mie model calculated dry particle backscatter or extinction and those fitted from humidograms.

[Figure]

**Figure S6.** Relationship between $\kappa_{\alpha355}$ and other 8 parameters.

**Table S1**. Determine coefficients (R2) between the 9 input parameters in Table 4.

| $R^2$ | $\kappa_{\alpha355}$ | $\kappa_{\alpha532}$ | $\kappa_{\beta355}$ | $\kappa_{\beta532}$ | $\kappa_{\beta1064}$ | $S_{a355}$ | $S_{a532}$ | $\mathring{a}_{\alpha355\&532}$ |
|---|---|---|---|---|---|---|---|---|
| $\kappa_{\alpha355}$ | = | = | = | = | = | = | = | = |
| $\kappa_{\alpha532}$ | 0.880 | = | = | = | = | = | = | = |
| $\kappa_{\beta355}$ | 0.411 | 0.321 | = | = | = | = | = | = |
| $\kappa_{\beta532}$ | 0.508 | 0.644 | 0.450 | = | = | = | = | = |
| $\kappa_{\beta1064}$ | 0.126 | 0.222 | 0.016 | 0.056 | = | = | = | = |
| $S_{a355}$ | 0.085 | 0.073 | 0.680 | 0.292 | 0.019 | = | = | = |
| $S_{a532}$ | 0.026 | 0.070 | 0.117 | 0.423 | 0.070 | 0.360 | = | = |
| $\mathring{a}_{\alpha355\&532}$ | 0.149 | 0.135 | 0.505 | 0.267 | 0.027 | 0.627 | 0.089 | = |
| $\mathring{a}_{\beta532\&1064}$ | 0.062 | 0.023 | 0.550 | 0.169 | 0.464 | 0.409 | 0.023 | 0.317 |

**S3 S6 Determine the tuning parameters for Random Forest model**

In this study, we use the Python module *RandomForestRegressor* from the Python Scikit-Learn library (http://scikit-learn.org/stable/modules/generated/sklearn.ensemble.RandomForestRegressor.html, last access: 18 December 2018) as the Random Forest (RF) model tool. The tuning parameters of the model are listed in Table S1S2. More detailed meanings about the setting values please refer to the user guide provided by the website.

The most import tuning parameter in the model is the number of trees in the forest (*n_estimators*). The influence of *n_estimators* on the accuracy of retrieved CCN number concentrations is tested. Here we use the same test method as introduced in Section 4.2 in the paper. The determination coefficients ($R^2$) and the mean absolute relative error (MARE) between theoretical calculated and retrieved CCN number concentrations with different *n_estimators* are shown in FigureFig. S3S7. The accuracy of the predictions increases as *n_estimators* grows bigger and are insensitive when *n_estimators* is bigger than 60. Considering computational and time cost, we finally set *n_estimators* to 100.

**Table S2S1**. Tuning parameters and their setting values of the Python module *RandomForestRegressor*.

| Parameter | Description | Values |
|---|---|---|
| *n_estimators* | The number of trees in the forest | 100 |
| *criterion* | The function to measure the quality of a split | "mse" |
| *max_features* | The number of features to consider when looking for the best split | "auto" |
| *max_depth* | The maximum depth of the tree | None |
| *min_samples_split* | The minimum number of samples required to split an internal node | 2 |
| *min_samples_leaf* | The minimum number of samples required to be at a leaf node | 1 |
| *min_weight_fraction_leaf* | The minimum weighted fraction of the sum total of weights (of all the input samples) required to be at a leaf node | 0 |
| *max_leaf_nodes* | Grow trees with *max_leaf_nodes* in best-first fashion | None |
| *min_impurity_decrease* | A node will be split if this split induces a decrease of the impurity greater than or equal to this value | 0 |

[Figure]

**Figure S73.** Influence of the number of trees in RF model on retrieving CCN number concentrations. Dependencies of tree numbers on **(a)** $R^2$ and **(b)** MARE between theoretical calculated CCN number concentrations and retrieved CCN number concentrations under different supersaturations.

[Figure]

**Figure S84.** Comparison of the theoretical calculated extinction-related CCN activation ratio at 532 nm and the model predicted extinction-related CCN activation ratios at 532 nm at supersaturations of **(a)** 0.20%, **(c)** 0.40%, and **(e)** 0.80%, and of the theoretical calculated CCN number concentrations and the retrieved CCN number concentrations at supersaturations of **(b)** 0.20%, **(d)** 0.40%, and **(f)** 0.80%. A total of 80575 pairs of data calculated from campaign C5 are used. The solid line is 1:1 line, and the dashed lines are 20% relative difference lines. Colors represent the relative density of the data points normalized by the maximum data density of each panel. The relative error showed in the figure is mean value ± one standard deviation.

[Figure]

**Figure S95.** Comparison of the theoretical calculated extinction-related CCN activation ratio at 532 nm and the model predicted extinction-related CCN activation ratios at 532 nm at supersaturations of **(a)** 0.07%, **(c)** 0.10%, and **(e)** 0.20%, and of the theoretical calculated CCN number concentrations and the retrieved CCN number concentrations at supersaturations of **(b)** 0.07%, **(d)** 0.10%, and **(f)** 0.20%. A total of 80575 pairs of data calculated from campaign C5 are used. The solid line is 1:1 line, and the dashed lines are 20% relative difference lines. Colors represent the relative density of the data points normalized by the maximum data density of each panel. The relative error showed in the figure is mean value $\pm$ one standard deviation.

---

## Author Comment (AC2) · 27 Apr 2019

Reply to Anonymous Referee #2

General

The paper deals with an important topic of atmospheric research: The retrieval of CCN from multiwavelength backscatter and extinction lidar observations. New aspects are integrated (e.g., using the increase in backscatter and extinction coefficient with relative humidity, humidogram approach). This is an excellent idea!

However, the paper has to my opinion almost no structure. Although one gets an idea how the method may work, a well-elaborated and well-structured description of the methodology is completely missing. A huge amount of information on microphysical and chemical particle particle properties, simulated related optical effects, dependence of optical effects from increasing humidity (growth factors), etc. and respective experimental data from field observations in the North China

Plain (NCP) are just accumulated in different sections, but the reader must already be an expert in this field to get an idea how all this may fit together.

So, the manuscript is far from being acceptable. To be short: to my opinion it cannot be published in the present version and must be rejected. In the following, I will give more detail so that the authors may get an idea how to change the structure of the manuscript and then resubmit it.

A manuscript with clear goals, clear structure, an easy-to-follow methodology, plus uncertainty analysis, and also some convincing but simple case studies are required.

**Response:**

Thanks for your suggestions. The structure of the paper has been rearranged and some additional explanations have been added according to your valuable suggestions. The retrieval algorithm is now presented at the beginning of Sect. 3 step by step. We also give explanations to the data we used in this study before we introduce the data. All of the comments and concerns raised by the referee have been explicitly replied point by point and incorporated into the revision.

An important issue is also: Please do not consider NCP aerosol only. It would be nice to have a general methodology that can be applied to anthropogenic haze and smoke, to dust aerosol, and may be to aerosols dominated by marine sea salt. All the required kappa values and growths factors are available in the literature. You do not need to focus on NCP aerosol!

**Response:**

Thanks for the valuable suggestions. We hope to propose a complete methodology as well. This paper is a very preliminary job. The main goal of our paper is introducing the idea of using backscatter and extinction humidograms to retrieve CCN and showing the significance of the information contained in the humidogram parameters. In this paper, we only use dataset measured in the NCP as an example to theoretically evaluate the feasibility of the method, because one of the biggest challenges to consider other types of aerosols is the acquisition of the size-resolved hygroscopicity data. Hygroscopicity (specially referred to here as $\kappa$) cannot be regarded as a size- independent parameter in most cases. The backscatter and extinction humidogram are influenced by hygroscopicity of all particles but the calculation of CCN number concentration is only controlled by hygroscopicity of small particles. Optical-equivalent bulk $\kappa$ is different from critical activation $\kappa$, and the relationship between them is found to be complex (Tao et al., 2018). Besides, the mixing state of particles is also an important parameter in simulating particle optical properties.

The purpose of using these data is to make the simulation closer to real situation in the atmosphere.

So we only focus on the aerosol in the NCP, and we believe that researchers who concern about other types of aerosols can repeat theoretical simulations with their own datasets. Also, we are trying to complete the algorithm as much as possible in future work.

Details

P3, L8-9: . . . The new method to retrieve CCN . . . is proposed based on kappa Koehler theory. . . ...
There are many examples of such 'isolated' phrases, not further explained. So, often the reader is confronted with a lot of information and then with the question: What do they want to say? How will that be used? Only experts know what the message behind probably is.

**Response:**

Thanks for pointing the problem out. We revised the sentence:
*'In this paper, a new method to retrieve CCN number concentrations for 3β+2α MWRL systems is proposed. Theoretical simulations are carried out to seek the relationship between CCN number concentrations and lidar-derived optical properties. The simulation implement κ-Köhler theory (Petters and Kreidenweis, 2007) to describe particle hygroscopic growth and activation process. Mie theory (Bohren and Huffman, 2007) is utilized to calculate particle backscatter and extinction coefficients from in situ measured aerosol microphysical and chemical properties.'*

Section 2: As a motivation it is mentioned: Section 2 introduces the measured and simulated data sets. Ok. But why do you introduce these data, why do you need these data, what is the goal, do you need them as input? All this is not explained. Nothing is clear. As a reader one wants to know the motivation for every section, before the content of the section is presented.

**Response:**

An introduction of the motivation for Sect. 2 has been added to the beginning of the section:
*'Since it is not easy to accumulate large datasets of simultaneous measurements of lidars and aircrafts, ground-measured aerosol microphysical and chemical data are used to simulate lidar-derived backscatter and extinction coefficients and corresponding CCN number concentrations. The simulations are based on κ-Köhler theory and Mie theory. The required datasets include: particle number size distribution (PNSD), black carbon (BC) mass concentrations ($m_{BC}$), mixing state of BC containing particles, and size-resolved hygroscopicity. The simulation results are used to establish and validate the new retrieval method.'*

And after reading of section 2 and all the subsequent sections it becomes more or less clear: This method obviously only holds for North China Plain (NCP) pollution aerosol. But is that then a useful method, for all the lidar scientists around the globe?
As a reader I expect a well elaborated algorithm applicable to all relevant aerosol types: What about pure marine conditions, what a about desert dust scenarios, what about forest fire smoke in the free troposphere? How does the method work in these cases? All this is not considered in this paper. Only NCP aerosol.

**Response:**

We agree that other aerosol types should be considered if the algorithm need to be applied globally. However, the main purpose of our paper is to introduce the idea of using humidograms to estimate CCN to lidar and other related researchers. Elaborating an algorithm is a long process and needs the contribution of the whole community. We sincerely hope to share the idea with an example of NCP

to other researchers through the paper and improve the algorithm together in the future.

The BC mixing state is introduced. . . . size resolved chemical composition all come from campaign
C2. . .. . . and after transforming the ambient wet aerodynamic diameters into dry volume-equivalent
diameters, size resolved kappa distribution were derived from measured size resolved chemical
composition. Twenty five typical size-resolved kappa distributions in the NCP are computed and
later used. . ... in the simulations. Who can follow? Why do you present all this? The methodlogy is
still not presented yet! The paper is to more than 50% just an acccumulation of information, like in
a laboratory book . . .. Section 2.2.
**Response:**
The derivation of the parameter of BC mixing state and size-resolved $\kappa$ distributions are detailed
described in the papers of Ma et al. (2012) and Liu et al. (2014) respectively. The use of these data
can improve the simulation. The mixing state of BC influence particle optical properties, and the $\kappa$
distribution, as is addressed in the previous response, is needed in the simulations of humidogram
and CCN activation. These valuable parameters are important in the method. All the data used in
this paper will be put in a repository and available to everyone if the paper is published.

P4, Eq3 is introduced, taken from another paper, D is introduced as particle/droplet radius (but it is
obviously the droplet radius). The equation contains: RH = 1 + SS with SS in %, so what is the unit
of RH?
**Response:**
Thanks for your suggestion. We have changed 'particle/droplet' to 'particle or droplet'. As for the
doubt about the unit of RH, we check the equation carefully and believe there is no problem about
the equation. Besides, there is no indication that the unit of SS is '%' in the manuscript.

P4, L29: Then Mie theory is used with all the parameters introduced before including the confusing
25 kappa distributions. It is impossible to check the simulations in detail. We are forced to read
another paper (Zhao et al, 2017). So, as a reader I am totally confused by the accumulation of all
the information from field campaigns, simulations, partly explained in another paper. All particles
are spherical which may make sense for North China pollution close to cloud base but what about
aerosol mixtures with a lot of mineral dust, practically 75% of the aerosol in China) probably
contains dust, what do we do in these cases?
**Response:**
The simulations are now introduced in detail in the Supplement (Please see Sect. S3 of the
Supplement). Assumptions, equations, and values of key parameters are presented.
We agree that mineral dust is an important component of aerosols in China. NCP is also influenced
by dust transported from North East China, especially in spring. However, limited by the knowledge
and data, dust is not considered in the current algorithm. When dust is the dominant component
(easily identified from polarization measurement), the algorithm is not applicable at present.
Considering the mixture of anthropogenic aerosols and mineral dust is a challenge, especially for
dust coated by anthropogenic components (e.g. sulfate or organic compounds). The optical
properties, hygroscopicity, and size-resolved number fraction of dust aerosols coated by
anthropogenic components still have lots of uncertainties. More measurements are needed to deal
with these cases.

**Response:**

In this part, we want to discuss the number and hygroscopic information of particles that can be gained from lidar measurements. Lidar can be used to retrieve CCN number concentration only if we can get particle number and hygroscopic information of CCN-related sizes. Size-resolved enhancement contributions of backscatter and extinction are discussed because we want to know hygroscopicity information of which size we can get from the humidograms of backscatter and extinction. The result shows that the humidograms of different parameters are sensitive to the hygroscopicity of different sizes. The oscillation in backscatter means the hygroscopic information in backscatter enhancement is complex and different from that in extinction enhancement. That explains why we use humidogram parameters ( $\kappa_\xi$ ) of all the $3\beta+2\alpha$ to indicate particle hygroscopicity. Size-dependent hygroscopicity is important to estimate CCN rather than a bulk hygroscopicity information, especially for different supersaturation conditions. One humidogram may indicate the bulk hygroscopicity, but it is the hygroscopicity of small particles that influences CCN number concentrations most. Spectrally dependent $\kappa_\xi$ can provide some information about the hygroscopicity of small particles. We have removed the 'oscillation' sentence and add the explanation:

*'Figure 3b also shows that different $\kappa_\xi$ is sensitive to the hygroscopicity of different size. Size-dependent hygroscopicity is important to estimate CCN rather than a bulk hygroscopicity information, especially for different supersaturation conditions. One humidogram may indicate the bulk hygroscopicity, but it is the hygroscopicity of small particles that influences CCN number concentrations most. Using $\kappa_\xi$ of all the $3\beta+2\alpha$ can provide some information about the hygroscopicity of small particles.'*

This part has been moved to Discussion (Section 4.1) with some extra explanation in the revised version.

**Response:**

Thanks for your suggestion to change the structure. The algorithm is now presented at the beginning of Sect. 3 and is described step by step as you suggested. We not only want to show the algorithm to researchers but also hope to introduce how we develop the algorithm and the scientific support
behind the method.
Now the methodology section starts, Sect. 3.3.
P7 L6: The equation provides the basic relationship between lidar information (backscatter and
extinction coefficient) and N-CCN.
P7 L4: you write backscatter/extinction coefficient, but I believe you want to write backscatter and
extinction ...., and not to use the ratio (backscatter/extinction). All this is confusing!
**Response:**
Thanks for the suggestion. We now use 'backscatter and extinction' or 'backscatter or extinction'
according to the meaning in the manuscript.
P7, Eq.7 and Eq.8, again new parameters are introduced, new discussion and uncertainty sources,
but no clear flow chart what to do with all this information in detail (step byr step).
P8, L4-6: Here the entire method is summarized within three lines! We need to study Figure 3 that
shows a flow chart. This is a SKTECH! . . . and helps to understand the method. But a clear set of
equations with all input parameters needed in the first step and all the output parameters, which are
again input for the next step and so on, is missing. All this is needed for five optical properties (3
beta and 2 ext). All this is not presented. What about uncertainties in the retrieval? How can we get
a convincing opinion on the potential (and especially the limits) if we have only Fig 3 and then the
correlation plots in Figure 4. Figures 6 and Figure 7 are useful. But we need to see the overall
concept (equations, including uncertainty computation approaches.) And we need it for other
aerosol types (dust, haze, marine..), not only for NCP aerosol which is of course an important aerosol
mixture.
**Response:**
Thank you for your advice. We have already expressed the retrieval step more clearly according to
your suggestion.
P8, L4-6 describes the last step of the algorithm, and we have added a word 'Finally' to avoid
misunderstandings.
We have tried to make a flow chart to contain all the details, but it would make the figure really
complicated. So we remain the flow chart unchanged, because it is concise and easy for the readers
to understand. For more detailed information, the algorithm is now described step by step in Sect.
3.1, and we believe it is now clear to the readers. The idea of our method is more valuable than the
details in the algorithm, and all the steps is adjustable if others want to use the idea to retrieve CCN
from lidars.
The uncertainties of the retrieval shown in Fig. 4 indicate the uncertainties from the concept of
method. It shows whether humidogram parameters, Ångström exponent, and lidar ratios can
describe the variation of the defined optical-related activation ratio. Uncertainties can also arise
from the assumption of the aerosol model, the $\kappa$-Köhler theory, and the Mie theory. All the
assumptions and theories are not exactly what the real world is. These uncertainties are evaluated
by lots of closure studies. In practical application, uncertainties also come from lidar and other
corresponding measurements. These uncertainties could be analyzed by Monte-Carlo method if the
uncertainties of all the measured value is known.
As for the limitations, we have already addressed in the summary in the public version of the manuscript (the second last paragraph):

[revised manuscript text omitted]

**S2 S3 Black carbon size distribution used in tTheoretical simulations of CCN, lidar backscatter, and extinction**

**S3.1 Aerosol model assumptions**

We assume the aerosol particles act as follows:

(1) Aerosol particles are spherical, which means the simulation results are not appropriate for mineral dust.

(2) Particles are partially externally mixed and partially core-shell mixed. Only two kinds of aerosols are contained: pure black carbon (BC) and BC coated by non-light-absorbing components. Note that if $r_{ext} = 1$, there exists pure BC and pure non-light-absorbing particles. $r_{ext}$ is defined with Eq. (2) in the paper.

(3) The shape of BC mass size distribution (BCMSD) remains unchanged and the amount is related to the total BC mass concentration ($m_{BC}$). The fixed distribution comes from the average BCMSD obtained from Berner impactor measurements (Ma et al., 2012) and is shown in Fig. S3.

[Figure]

**Figure S32.** Normalized size distribution of black carbon (BC) mass/volume concentration. The distribution is the average black carbon mass concentration obtained from Berner impactor measurements (Ma et al., 2012) and is normalized by the maximum value of the distribution.

(4) $r_{ext}$ is uniform among different particle sizes. Accordingly, number concentrations of externally mixed BC ($N_{ext}$) can easily be calculated from BCMSD and $r_{ext}$:

$$N_{ext}(D) = \frac{r_{ext} \cdot m_{BC}(D)}{\frac{\pi}{6} D^3 \cdot \rho_{BC}}, \tag{S1}$$

where $D$ is diameter of the particle, $\rho_{BC}$ is the density of BC. In this study, $\rho_{BC}$ is assumed to be 1.5 g/cm³, which is also the density when retrieving $r_{ext}$.

(5) For each particle size, the diameters of BC cores ($D_{core}$) are the same and can be derived using the following equation:

$$D_{core}(D) = \sqrt[3]{\frac{6(1-r_{ext}) \cdot m_{BC}(D)}{\pi \rho_{BC} \cdot [N(D) - N_{ext}(D)]}}, \tag{S2}$$

where $N(D)$ represents particle number size distribution (PNSD).

(6) The size-resolved $\kappa$ distributions represent the bulk hygroscopicity of core-shell mixed particles, and externally mixed BC particles do not take up water.

All these assumptions above are strong and, to some extent, inconsistent with the reality, but are certified to be reasonable for calculating aerosol optical properties. Plenty of works on aerosol optical closure studies (Ma et al., 2011;Ma et al., 2012) and aerosol optical simulations (Kuang et al., 2017;Kuang et al., 2018;Zhao et al., 2018) have been carried out with these aerosol model assumptions. In particular, Zhao et al. (2017) use the aerosol model to simulate lidar backscatter and extinction under different relative humidity (RH) conditions.

**S3.2 Calculations of CCN number concentrations using $\kappa$-Köhler theroy**

According to $\kappa$-Köhler theory, CCN number concentrations at a specific supersaturation level can be calculated by PNSD and size-resolved $\kappa$ distribution. Based on Eq. (3), the critical supersaturation ratio required to activate a particle is decided by corresponding $\kappa$ and $D_{\text{dry}}$. In other word, we can get a critical activation dry diameter $D_c$ with a given supersaturation ratio and size-resolved $\kappa$ distribution. Then CCN number concentration $N_{\text{CCN}}(SS)$ thereby can be calculated with Eq. (S3):

$$N_{\text{CCN}}(SS) = \int_{D_c(SS)}^{D_{\max}} [N(D) - N_{\text{ext}}(D)]\, dD\,, \tag{S3}$$

where $D_{\max}$ correspond to the upper bounds of the measured PNSD. Note that we regard externally mixed pure BC as non-hygroscopic particles, so $r_{\text{ext}}$ and $m_{\text{BC}}$ should also be involved to calculate $N_{\text{ext}}$, which needs to be subtracted. Otherwise, $N_{\text{CCN}}$ will be overestimated.

**S3.3 Calculations of particle backscatter and extinction coefficients at different RH using $\kappa$-Köhler theroy and Mie theory**

We use a modified BHMIE Fortran code and a modified BHCOAT Fortran code to calculate optical properties of homogeneous spherical particles and coated spherical particles, respectively. For a homogeneous spherical particle (i.e. externally mixed BC in this study), BHMIE can calculate particle scattering and extinction efficiency ($Q_{\text{sca}}$ and $Q_{\text{ext}}$) and scattering phase function with given light wavelength, particle diameter, and complex refractive index. For a coated spherical particle (i.e. core-shell mixed particle), diameters and complex refractive indices of both core and shell are needed in BHCOAT. The particle backscatter and extinction coefficients we need are derived from $Q_{\text{sca}}$, $Q_{\text{ext}}$, and scattering phase function at 180° $P(\pi)$:

$$\alpha = \sum_i \left[ \int_{D_{\min}}^{D_{\max}} \frac{1}{4} D^2 Q_{\text{ext}}(D, i) N(D, i)\, dD \right], \tag{S4}$$

$$\beta = \sum_i \left[ \int_{D_{\min}}^{D_{\max}} \frac{1}{16} D^2 Q_{\text{sca}}(D, i) P(\pi, D, i) N(D, i)\, dD \right], \tag{S5}$$

where $D_{\min}$ and $D_{\max}$ correspond respectively to the lower and upper bounds of the measured PNSD, and the index $i$ indicates the mixing state of particles, i.e. external or core-shell in this paper. Polarization of lidar emitted laser is neglected.

Complex refractive indices are essential for Mie scattering calculation. Aerosol complex refractive indices are related to chemical components, morphology, and wavelengths of light (Cotterell et al., 2017). Both real part and imaginary part of refractive indices vary a lot in real ambient environment (Shettle and Fenn, 1979). Wavelength dependency of refractive indices at wavelengths of 355 nm, 532 nm, and 1064 nm are not significant except for brown carbon (Shettle and Fenn, 1979;Bond et al., 2013). Neglecting the effect of brown carbon in this study, we simply assume that complex refractive indices of corresponding components do not change with wavelengths. The refractive indices of BC and non-light-absorbing component (shell) are set to be $1.8+0.54i$ (Ma et al., 2012) and $1.53+10^{-7}i$ (Wex et al., 2002), respectively.

Concerning aerosol hygroscopicity for core-shell mixed particles, diameters of BC cores $D_{core}$ remain unchanged, and particle diameters $D$ at different RH can be calculated with Eq. (3). The refractive index of the swelling shell ($\widetilde{m}_{shell}$) is calculated following the volume mixing law (Hanel, 1968):

$$\widetilde{m}_{shell} = f_{solute} \cdot \widetilde{m}_{solute} + (1 - f_{solute}) \cdot \widetilde{m}_{water}, \tag{S6}$$

where $\widetilde{m}_{solute}$ is the refractive index of solute (i.e. $1.53+10^{-7}i$ in this study), $\widetilde{m}_{water}$ is the refractive index of pure water ($1.33+10^{-7}i$), and $f_{solute}$ is the solute volume fraction of the in solution (shell), which is determined by Eq. (S7):

$$f_{solute} = \frac{D_{dry}^3 - D_{core}^3}{D^3 - D_{core}^3}. \tag{S7}$$

[Figure]

**Figure S4.** Example of humidogram fitting using different functions. The example is calculated with one set of PNSD, BC, $r_{ext}$, and size-resolved $\kappa$ distribution. The dots represent Mie model simulations, and the dots in red (within RH range of 60-90%) are used to fit parameterization lines. The blue line is the result of $\gamma$-equation, and the green line represents the result of $\kappa$-equation.

[Figure]

**Figure S5.** Comparison between Mie model calculated dry particle backscatter or extinction and those fitted from humidograms.

[Figure]

**Figure S6.** Relationship between $\kappa_{\alpha355}$ and other 8 parameters.

**Table S1**. Determine coefficients (R2) between the 9 input parameters in Table 4.

| $R^2$ | $\kappa_{\alpha355}$ | $\kappa_{\alpha532}$ | $\kappa_{\beta355}$ | $\kappa_{\beta532}$ | $\kappa_{\beta1064}$ | $S_{a355}$ | $S_{a532}$ | $\mathring{a}_{\alpha355\&532}$ |
|---|---|---|---|---|---|---|---|---|
| $\kappa_{\alpha355}$ | = | = | = | = | = | = | = | = |
| $\kappa_{\alpha532}$ | 0.880 | = | = | = | = | = | = | = |
| $\kappa_{\beta355}$ | 0.411 | 0.321 | = | = | = | = | = | = |
| $\kappa_{\beta532}$ | 0.508 | 0.644 | 0.450 | = | = | = | = | = |
| $\kappa_{\beta1064}$ | 0.126 | 0.222 | 0.016 | 0.056 | = | = | = | = |
| $S_{a355}$ | 0.085 | 0.073 | 0.680 | 0.292 | 0.019 | = | = | = |
| $S_{a532}$ | 0.026 | 0.070 | 0.117 | 0.423 | 0.070 | 0.360 | = | = |
| $\mathring{a}_{\alpha355\&532}$ | 0.149 | 0.135 | 0.505 | 0.267 | 0.027 | 0.627 | 0.089 | = |
| $\mathring{a}_{\beta532\&1064}$ | 0.062 | 0.023 | 0.550 | 0.169 | 0.464 | 0.409 | 0.023 | 0.317 |

**S3 S6 Determine the tuning parameters for Random Forest model**

In this study, we use the Python module *RandomForestRegressor* from the Python Scikit-Learn library (http://scikit-learn.org/stable/modules/generated/sklearn.ensemble.RandomForestRegressor.html, last access: 18 December 2018) as the Random Forest (RF) model tool. The tuning parameters of the model are listed in Table S1 S2. More detailed meanings about the setting values please refer to the user guide provided by the website.

The most import tuning parameter in the model is the number of trees in the forest (*n_estimators*). The influence of *n_estimators* on the accuracy of retrieved CCN number concentrations is tested. Here we use the same test method as introduced in Section 4.2 in the paper. The determination coefficients ($R^2$) and the mean absolute relative error (MARE) between theoretical calculated and retrieved CCN number concentrations with different *n_estimators* are shown in Figure Fig.

S3 7. The accuracy of the predictions increases as *n_estimators* grows bigger and are insensitive when *n_estimators* is bigger than 60. Considering computational and time cost, we finally set *n_estimators* to 100.

**Table S2 1**. Tuning parameters and their setting values of the Python module *RandomForestRegressor*.

| Parameter | Description | Values |
|---|---|---|
| *n_estimators* | The number of trees in the forest | 100 |
| *criterion* | The function to measure the quality of a split | "mse" |
| *max_features* | The number of features to consider when looking for the best split | "auto" |
| *max_depth* | The maximum depth of the tree | None |
| *min_samples_split* | The minimum number of samples required to split an internal node | 2 |
| *min_samples_leaf* | The minimum number of samples required to be at a leaf node | 1 |
| *min_weight_fraction_leaf* | The minimum weighted fraction of the sum total of weights (of all the input samples) required to be at a leaf node | 0 |
| *max_leaf_nodes* | Grow trees with *max_leaf_nodes* in best-first fashion | None |
| *min_impurity_decrease* | A node will be split if this split induces a decrease of the impurity greater than or equal to this value | 0 |

[Figure]

**Figure S73.** Influence of the number of trees in RF model on retrieving CCN number concentrations. Dependencies of tree numbers on **(a)** $R^2$ and **(b)** MARE between theoretical calculated CCN number concentrations and retrieved CCN number concentrations under different supersaturations.

[Figure]

**Figure S8.** Comparison of the theoretical calculated extinction-related CCN activation ratio at 532 nm and the model predicted extinction-related CCN activation ratios at 532 nm at supersaturations of **(a)** 0.20%, **(c)** 0.40%, and **(e)** 0.80%, and of the theoretical calculated CCN number concentrations and the retrieved CCN number concentrations at supersaturations of **(b)** 0.20%, **(d)** 0.40%, and **(f)** 0.80%. A total of 80575 pairs of data calculated from campaign C5 are used. The solid line is 1:1 line, and the dashed lines are 20% relative difference lines. Colors represent the relative density of the data points normalized by the maximum data density of each panel. The relative error showed in the figure is mean value ± one standard deviation.

[Figure]

**Figure S9 5.** Comparison of the theoretical calculated extinction-related CCN activation ratio at 532 nm and the model predicted extinction-related CCN activation ratios at 532 nm at supersaturations of **(a)** 0.07%, **(c)** 0.10%, and **(e)** 0.20%, and of the theoretical calculated CCN number concentrations and the retrieved CCN number concentrations at supersaturations of **(b)** 0.07%, **(d)** 0.10%, and **(f)** 0.20%. A total of 80575 pairs of data calculated from campaign C5 are used. The solid line is 1:1 line, and the dashed lines are 20% relative difference lines. Colors represent the relative density of the data points normalized by the maximum data density of each panel. The relative error showed in the figure is mean value ± one standard deviation.

---

## Author Response (AR2)

Dear editor,

We really appreciated the reviewers for their patience and nice suggestions. The manuscript is
revised according to their suggestions. Please see the following responses.

Yours,

Authors

**Reply to Anonymous Referee #1**

I am satisfied by the authors' responses to my review and I recommend this paper be accepted for
publication in AMT.
I do have one additional minor correction regarding the caption in Fig. 4. In the last sentence please
change "showed" to "shown".

**Response:**

Revised.

**Reply to Anonymous Referee #2**

The manuscript is greatly improved. However, it is still not easy to read and one has to be an expert
in the field to understand the approach. All in all, it is a very complex and complicated paper.
Some recommendations:
One should discuss the two recently published CCN-lidar approaches (Lv et al., JGR 2018,
Mamouri and Ansmann, ACP 2016) in more detail in the introduction. What is the idea in these two
foregoing papers? How do the ideas presented here differ from the foregoing methods.
The paper of Mamouri and Ansmann is based on single-wavelength polarization lidar (and not on
multiwavelength lidar), and applicable to marine, dust, and pollution aerosol types.

**Response:**

Additional discussions about the two recent papers were added to the introduction (page 2, line 15-
23; page 2, line 29-34 for the marked-up version).

*Mamouri and Ansmann (2016) investigate the potential of single-wavelength polarization lidar to*
*retrieval CCN for three aerosol types (desert, non-desert continental, and marine). The polarization*
*lidar can separate desert and non-desert by means of the particle linear depolarization ratio. Based*
*on datasets from multiyear Aerosol Robotic Network (AERONET) observations, valid relationships*
*are found between particle extinction coefficients and number concentrations of particles with dry*
*radius larger than 50 nm (for non-desert and marine) and 100 nm (for desert). CCN concentrations*
*at different supersaturations are parameterized with the particle number concentration derived from*
*extinction profiles according to aerosol types. The consideration of the hygroscopicity of ambient*
*particles is empirical. Besides, single-wavelength lidar also lacks of sufficient information to*
*quantify particle number concentration, which will bring large uncertainty on CCN retrieval.*
*Lv et al. (2018) build a look-up table based on AERONET datasets to retrieve particle number size*
*distributions from backscatter and extinction profiles. Then assumed activation critical diameters*
*according to aerosol type classification together with the retrieved optical-equivalent particle size*
*distributions are utilized to calculate CCN concentrations. It is worth noting that most of the*
*foregoing methods implement crude particle type classification to determent particle hygroscopicity.*
The statement of the difference of our method is also added (page 3, line 21-28 for the marked-up version).

*Different from the foregoing approaches which use AERONET datasets, we use in situ measured*
*microphysical and chemical data in this study. Theoretical simulations based on in situ*
*measurements are carried out to seek the relationship between CCN number concentrations and*
*lidar-derived optical properties. The simulation implements κ-Köhler theory (Petters and*
*Kreidenweis, 2007) to describe particle hygroscopic growth and activation process. Mie theory*
*(Bohren and Huffman, 2007) is utilized to calculate particle backscatter and extinction coefficients*
*from in situ measured aerosol microphysical and chemical properties. The enhancements of*
*backscatter and extinction with RH are implemented to quantify particle hygroscopicity instead of*
*using empirical estimation according to aerosol type classification.*

One should also clearly state …. right in the beginning and thus in the introduction: The new method
is only applicable to well mixed aerosol layers with large changes in the relative humidity, and thus
only in the case of the planetary boundary layer. Furthermore, the method works only in the absence
of dust (non spherical particles).

**Response:**
The statement was added to the introduction. The method is not limited to the cases with large
changes in RH, because the RH range is flexible. For different cases with different RH ranges,
researchers only need to change the input data (parameters fitted from different RH ranges) for the
random forest model. The RH range in the paper is just an example. Also, well-mixed layers are
commonly found in the planetary boundary layer (PBL), but they can appear in the free troposphere,
especially for cumulus above the PBL. The following statement was added (page 3, line 28-31 for
the marked-up version):

*The new method is applicable to well-mixed aerosol layers. We take datasets in the North China*
*Plain (NCP) as an example of this method. The NCP is influenced by heavy and complex pollution*
*which shows strong characteristics of continental aerosols. Mineral dust and marine particles are*
*not considered in this study.*

You could compare your AR values with published ones in Mamouri and Ansmann 2016 and
Shinozuka, Y., et al. : The relationship between cloud condensation nuclei (CCN) concentration and
light extinction of dried particles: indications of underlying aerosol processes and implications for
satellite-based CCN estimates, Atmos. Chem. Phys., 15, 7585-7604, https://doi.org/10.5194/acp-
15-7585-2015, 2015.

**Response:**
Mamouri and Ansmann (2016) describe the relationship between extinction coefficient at 532 nm
and number concentrations of particle with radius larger than 60 ($n_{60}$) and 100 nm ($n_{100}$) (see Figure
R1). The extinction and number concentrations are all in ambient condition. According to Mamouri
and Ansmann (2016), $n_{60}$ represents $n_{50}$ at dry condition and $n_{100}$ is for desert dust which do not take
up water. From Fig. 3(a) in our paper, $n_{60}$ can be compared with CCN concentrations at
supersaturation of 0.20%, and $n_{100}$ can be compared with CCN concentrations at supersaturation of
0.10%. Shinozuka et al. (2015) describe the relationship between dry extinction coefficient at 532
nm and CCN number concentrations at supersaturation of 0.40% (see Figure R2). Figure R3 gives
the results using our simulated data. As there are so many data points, 1000 pairs of data are selected
randomly to plot in Fig. R2. The result agrees well with their results in magnitude.

[Figure]

**Figure R1.** Figure 4(a) (left) and Figure 5(a) (right) in the paper of Mamouri and Ansmann (2016).

[Figure]

**Figure R2.** Figure 1 in the paper of Shinozuka et al. (2015)

[Figure]

**Figure R3.** Relationship between dry extinction coefficient at 532 nm and CCN number concentrations at supersaturations of 0.10% (red), 0.20% (green), and 0.40% (grey).

If your RH is retrieved by using microwave radiometer (no water vapour Raman lidar) then relative uncertainties in the range of 10-20% are much more realistic as your 5% (in all your simulations).

**Response:**

We added additional two tests of random error considering RH uncertainty of 10% and 20%. Table

6 (Table R1) and Figure 8 (Figure R4) are accordingly updated. Description and discussion in the manuscript are also renewed.

**Table R1.** Mean and one standard deviation (std) values (mean ± std) of relative errors in retrieved

CCN number concentrations at different supersaturations with different random error conditions.

The uncertainty of backscatter and extinction coefficients of all the tests is 10%, and the uncertainties of relative humidity are 5%, 10%, and 20%, respectively.

| Supersaturation ratio | Random error (10% for backscatter and extinction) | | |
|---|---|---|---|
| | Error of relative humidity | | |
| | **5%** | **10%** | **20%** |
| **0.07%** | $-4.1\% \pm 21.8\%$ | $0.2\% \pm 23.4\%$ | $0.7\% \pm 22.6\%$ |
| **0.10%** | $-1.5\% \pm 23.4\%$ | $-2.8\% \pm 24.0\%$ | $-2.5\% \pm 21.2\%$ |
| **0.20%** | $-1.2\% \pm 27.8\%$ | $-9.1\% \pm 26.3\%$ | $-5.2\% \pm 18.0\%$ |

[Figure]

**Figure R4.** Relative errors in fitted and calculated parameters with 10% random errors for backscatter and extinction and 5% (blue), 10% (orange), and 20%(green) random error for relative humidity. The dots are the median values, and the error bars denote the 5th and 95th percentiles.

The dashed red line marks the position of zero.

Figure4: very confusing y-axis and x-axis parameters:

For 'model-predicted extinction-related CCN activation ratio…'

may be simply use 'AR applied in the retrieval' or simply 'retrieval AR'

and for

'theoretical calculated extinction-related CCN activation ratio…'

take 'true AR' or 'modelled AR'

**Response:**

Revised (see Figure R5).

**Figure R5.** Comparison of the theoretical calculated extinction-related CCN activation ratio at 532 nm (true AR) and the model predicted extinction-related CCN activation ratios at 532 nm (retrieved AR) at supersaturations of **(a)** 0.07%, **(c)** 0.10%, and **(e)** 0.20%, and of the theoretical calculated CCN number concentrations (true CCN number concentration) and the retrieved CCN number concentrations at supersaturations of **(b)** 0.07%, **(d)** 0.10%, and **(f)** 0.20%. A total of 80575 pairs of data calculated from campaign C5 are used. The solid line is 1:1 line, and the dashed lines are 20% relative difference lines. Colors represent the relative density of the data points normalized by the maximum data density of each panel. The relative error shown in the figure is mean value ± one standard deviation.

[revised manuscript text omitted]